**Modeling the timing of Patagonian Ice Sheet retreat in the Chilean Lake District from 22-**
2                                              **10 ka**

4                        Joshua Cuzzone[1], Matias Romero[2], Shaun A. Marcott[2]
[1]Joint Institute for Regional Earth System Science and Engineering, University of California, Los
Angeles
[2]Department of Geoscience, University of Wisconsin, Madison
*Correspondence to*: Joshua K. Cuzzone (Joshua.K.Cuzzone@jpl.nasa.gov)
**Abstract**
Studying the retreat of the Patagonian Ice Sheet (PIS) during the last deglaciation represents an
important opportunity to understand how ice sheets outside the polar regions have responded to
deglacial changes in temperature and large-scale atmospheric circulation. At the northernmost
extension of the PIS during the last glacial maximum (LGM), the Chilean Lake District (CLD)
was influenced by the southern westerly winds (SWW), which strongly modulated the hydrologic
and heat budget of the region. Despite progress in constraining the nature and timing of deglacial
ice retreat across this area, considerable uncertainty in the glacial history still exists due to a lack
of geologic constraints on past ice margin change. Where the glacial chronology is lacking, ice
sheet models can provide important insight into our understanding of the characteristics and drivers
of deglacial ice retreat. Here we apply the Ice Sheet and Sea-level System Model (ISSM) to
simulate the LGM and last deglacial ice history of the PIS across the CLD at high spatial resolution
(450 meters). We present a transient simulation of ice margin change across the last deglaciation
using climate inputs from the CCSM3 Trace-21ka experiment.  At the LGM, the simulated ice
extent across the CLD agrees well with the most comprehensive reconstruction of PIS ice history
(PATICE).  Coincident with deglacial warming, ice retreat ensues after 19ka, with largescale ice
retreat occurring across the CLD between 18 and 16.5 ka.  By 17 ka the northern portion of the
CLD becomes ice free, and by 15 ka, ice only persists at high elevations as mountain glaciers and
small ice caps.  Our simulated ice history agrees well with PATICE for early deglacial ice retreat
but diverges at and after 15 ka, where the geologic reconstruction suggests persistence of an ice
cap across the southern CLD until 10 ka.  However, given the high uncertainty in the geologic
reconstruction of the PIS across the CLD during the later deglaciation, this work emphasizes a
need for improved geologic constraints on past ice margin change.  While deglacial warming drove
the ice retreat across this region, sensitivity tests reveal that modest variations in wintertime
precipitation (~10%) can modulate the pacing of ice retreat by up to 2 ka, which has implications
when comparing simulated outputs of ice margin change to geologic reconstructions.  While we
find that TraCE-21ka simulates large-scale changes in the SWW across the CLD that are consistent
with regional paleoclimate reconstructions, the magnitude of the simulated precipitation changes
is smaller than what is found in proxy records.  From our sensitivity analysis we can deduce that
larger anomalies in precipitation as found in paleoclimate proxies may have had a large impact on
modulating deglacial ice retreat, highlighting an additional need to better constrain the deglacial
change in the strength, position, and extent of the SWW as it relates to understanding the drivers
of deglacial PIS behavior.

## 1 Introduction

During the Last glacial maximum (LGM), the Patagonian Ice Sheet (PIS) covered the Andes mountains from 38°S to 55°S, with an estimated sea-level equivalent ice volume of 1.5 meters (Davies et al., 2020). At the northernmost extent of the PIS, across an area presently known as the Chilean Lake District (CLD: 37°S-41.5°S), the LGM to deglacial ice behavior and related climate forcings has been a subject of historical interest (Mercer, 1972; Porter, 1981; Lowell et al., 1995; Andersen et al., 1999; Denton et al., 1999; Glasser et al., 2008, Moreno et al., 2015; Kilian and Lamy, 2012; Lamy et al., 2010), and have served as important constraints towards understanding the drivers of ice sheet change across centennial to millennial timescales. Currently, PATICE (Davies et al., 2020) serves as the latest and most complete reconstruction of the entire PIS during the LGM and last deglaciation. Across the CLD (Figure 1), the LGM ice limits are only well constrained by terminal moraines in the southwest and western margins (Denton et al., 1999; Glasser et al., 2008, Moreno et al., 2015). However, due to a lack of geomorphological and geochronologic constraints on ice margin change following the LGM, the reconstructed deglaciation remains highly uncertain.

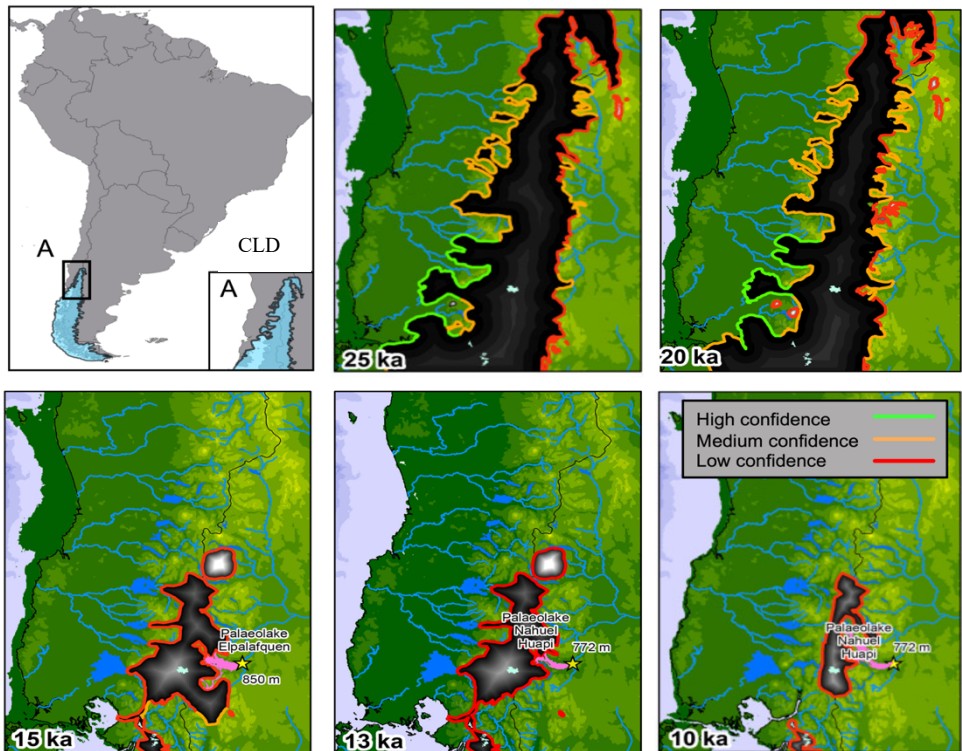

Figure 1. Location of the study area across the Chilean Lake District (CLD; Upper Left Panel). The reconstructed ice extent from PATICE for the PIS across the CLD at 25 ka, 20 ka, 15 ka, 13 ka, and 10 ka are taken from Davies et al., 2020. The color of the line marking the reconstructed ice extent corresponds to the confidence in the reconstruction as described in section 3.3.

While deglacial warming is a primary driver of ice retreat across the CLD, evidence suggests that variations in precipitation patterns influenced the timing and magnitude of this retreat (Moreno et al., 1999; Rojas et al., 2009). The wintertime climate across South America is strongly influenced by the southern annular mode (SAM; Hartmann and Lo, 1998), for which its phase and strength is regulated by changes in the difference of zonal mean sea-level pressure between mid (40°S) and

high latitudes (65°S). The SAM in turn modulates the strength and position of the southern
westerly winds (SWW) over decadal to multi-centennial timescales, which exert a large control on
the synoptic scale hydrologic and heat budget (Garreaud et al., 2013). During the LGM and last
deglaciation, paleoclimate data indicates that the position, strength, and extent of the SWW varied
latitudinally, migrating southward during warmer intervals and northward during cooler intervals,
ultimately altering overall ice sheet mass balance (Mercer, 1972; Denton et al., 1999; Lamy et al.,
2010; Kilian and Lamy, 2012; Boex et al., 2013). Terrestrial paleoclimate proxies that indicate
that the CLD was wetter during the LGM and early deglaciation have been used to support the
idea that the SWW migrated northward of 41°S across the CLD (Moreno et al., 1999; Moreno et
al., 2015; Moreno and Videla, 2018; Diaz et al., 2023). Additionally, these proxies indicate a
switch from hyper humid to humid conditions around 17,300 cal yr BP, which was inferred by
Moreno et al. (2015) to indicate the poleward migration of the SWW south of the CLD.
However, inferring changes in the SWW across the last deglaciation from paleoclimate proxies
can be problematic as outlined by Kohfeld et al. (2013) who compiled an extensive dataset of
paleoclimate archives that record changes in moisture, precipitation-evaporation balance, ice
accumulation, runoff and precipitation, dust deposition, and marine indicators of sea surface
temperature, ocean fronts, and biologic productivity. Kohfeld et al. (2013) conclude that
environmental changes inferred from existing paleoclimate data could be potentially explained by
a range of plausible scenarios for the state and change of the SWW during the LGM and last
deglaciation, such as a strengthening, poleward or equatorward migration, or no change. Climate
model results from Sime et al. (2013) indicate that the reconstructed changes in moisture from
Kohfeld et al. (2013) can be simulated well without invoking large shifts or changes in strength to
the SWW. This discrepancy also exists amongst climate models which diverge on whether the
LGM SWW was shifted equatorward or poleward, and was stronger or weaker than present day
(Togweiler et al., 2006; Menviel et al., 2008; Rojas et al., 2009; Rojas et al., 2013; Sime et al.,
2013; Jiang et al., 2020). Therefore, from paleoclimate proxies and climate models, we still do
not have a firm understanding of how the SWW may have changed during the last deglaciation,
and how these variations may have influenced the deglaciation of the PIS.
Early paleo ice sheet modelling experiments across the PIS have focused on evaluating the
relationship between the simulated LGM ice sheet geometry in response to spatially uniform
temperature change (Hulton et al., 2002; Sugden et al., 2002; Hubbard et al., 2005). While these
early simulations provided constraints on PIS areal extent, ice volume, and sensitivity to LGM
temperature depressions, spatially varying temperature and precipitation were not considered.
Recently, Yan et al. (2022) simulated the PIS behavior at the LGM using an ensemble of climate
model output from the Paleoclimate Modelling Intercomparison Project (PMIP4; Kageyama et al.,
2021). Results best matching the empirical reconstructions from PATICE (Davies et al., 2020)
suggest that reduction in temperature was likely the main driver of PIS LGM extent, although the
authors found that variation in regional LGM precipitation anomaly can have large impacts on the
simulated ice sheet geometry. This evidence is supported by recent glacier modelling across the
northeastern Patagonian Andes which suggests that increases in precipitation during the
termination of the LGM are necessary to achieve modeled fit with reconstructed glacier extent
(Muir et al., 2023; Leger et al., 2021b). Additionally, Martin et al. (2022) found that precipitation
greater than present day is needed to explain late glacial and Holocene ice readvance of the Monte
San Lorenzo ice cap, lying to the southeast of the current Northern Patagonian Ice Field. These
regional studies therefore provide further evidence that late glacial and deglacial variability in
precipitation, perhaps driven by changes in the SWW, influenced PIS retreat and readvance over
numerous timescales.
To advance our understanding of the last glacial and deglacial ice behavior across the CLD, we
use a numerical ice sheet model to simulate the LGM ice geometry and deglacial ice retreat using
transiently evolving boundary conditions from a climate model simulation of the last 21,000 years
(TraCE-21ka; Liu et al., 2009; He et al., 2013) which simulates large scale variability in the
strength and position of the SWW (Jiang and Yan, 2020). Because there is a lack of transiently
evolving ice sheet model simulations of the PIS across the last deglaciation, our aim is to provide
possible constraints on the nature of ice retreat across the CLD region, from which the
reconstructions (PATICE; Davies et al., 2020) are uncertain. Also, by assessing the sensitivity of
our ice sheet experiments to a range of climatic boundary conditions, we aim to provide additional
insight into the dominant climatic controls on the deglacial evolution of the PIS in the CLD region.
**2 Methods: Model description and setup**
**2.1 Ice sheet model**
In order to simulate the ice margin migration across the CLD during the LGM and last deglaciation,
we use the Ice Sheet and Sea-level System Model (ISSM), a thermomechanical finite-element ice
sheet model (Larour et al., 2012). Because of the high topographic relief across the CLD and
associated impact on ice flow, we use a higher-order approximation to solve the momentum
balance equations (Dias dos Santos et al., 2022). This ice flow approximation is a depth-integrated
formulation of the higher-order approximation of Blatter (1995) and Pattyn (2003), which allows
for an improved representation of ice flow compared with more traditional approaches in paleo-
ice flow modelling (e.g., Shallow Ice Approximation or hybrid approaches; Hubbard et al., 2005;
Leger et al., 2021b; Yan et al., 2022), while allowing for reasonable computational efficiency. Our
model domain comprises the northernmost LGM extent of the PIS across the CLD, extending
beyond the LGM ice extent reconstructed from Davies et al. (2020) and ends along the northern
shore of the Golfo de Ancud (Figure 2).
We rely on anisotropic mesh adaptation to create a non-uniform model mesh that varies based
upon gradients in bedrock topography from the General Bathymetric Chart of the Oceans
(GEBCO; GEBCO Bathymetric Compilation Group, 2021), a terrain model for ocean and land.
For the land component, the GEBCO model uses version 2.2 of the Surface Radar Topography
Mission data (SRTM15_plus; Tozer et al., 2019), to create a 15 arc second gridded output of terrain
elevation relative to sea level. Our ice sheet model horizontal mesh resolution varies from 3 km
in areas of low bedrock relief to 450 meters in areas where gradients in the bedrock topography is
high and comprises 40,000 model elements. We impose no boundary conditions of ice flow and
thickness at the southern extent of our model domain. Due to the north-south nature of the
simulated ice divide during the last deglaciation (see Figure 4), inflow from the south and into our
model domain is minimal and was found to not impact our results.


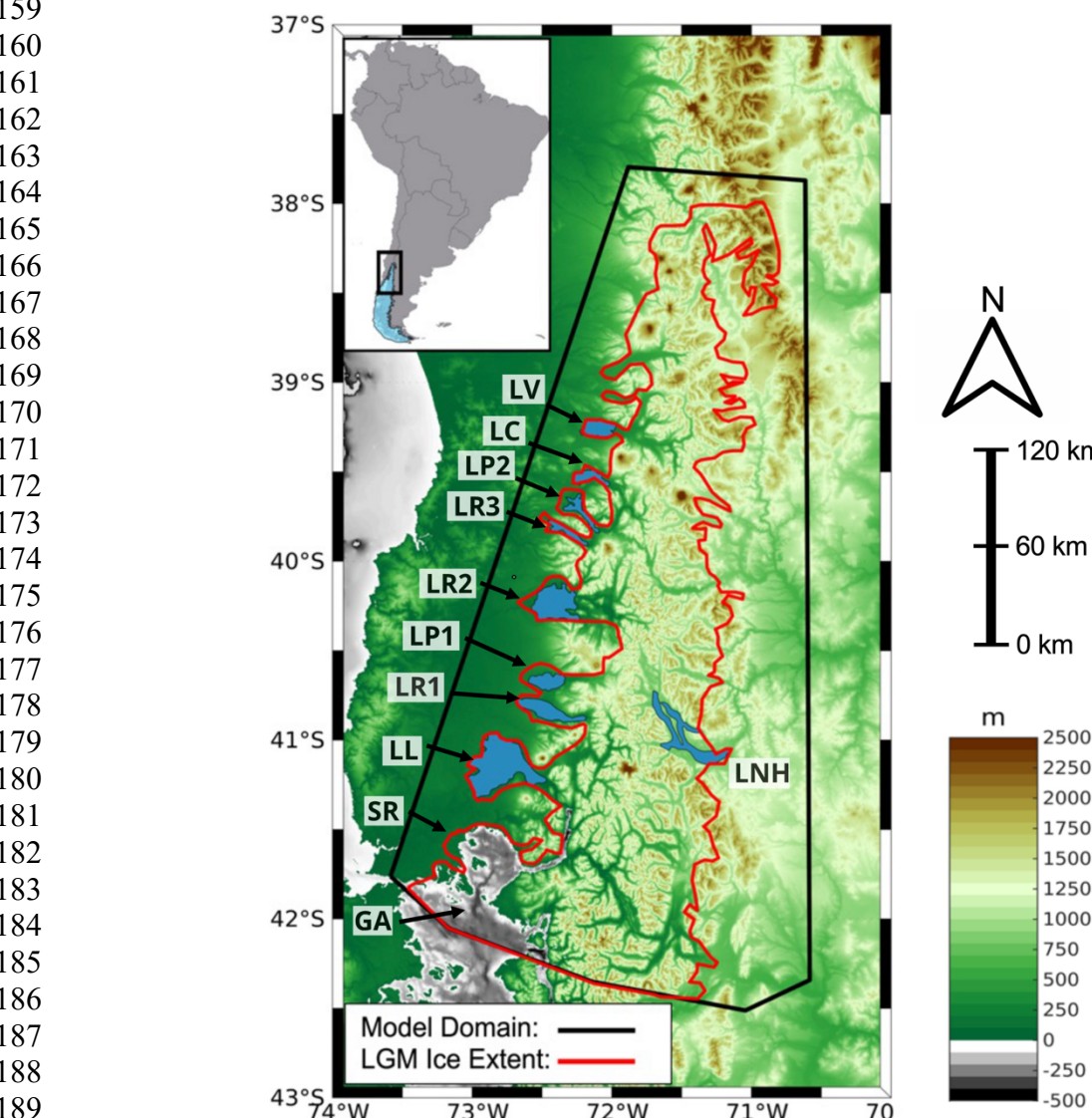

Figure 2. Bedrock topography for our study area (meters). Our model domain (shown as the black line), encompasses the reconstructed LGM ice limit (shown in red) from PATICE (Davies et al., 2020). Present day lakes are shown in blue, with abbreviated names as: SR (Seno de Reloncaví), GA (Golfo de Ancud), LL (Lago Llanquihue), LR1 (Lago Rupanco), LP1 (Lago Puyehue), LR2 (Lago Ranco), LR3 (Lago Riñihue), LP2 (Lago Panguipulli), LC (Lago Calafquén), LV (Lago Villarica), LNH (Lago Nahuel Huapi).

Although geomorphological evidence suggests that while southernmost glaciers across the PIS
may have been temperate with warm based conditions during the LGM, there may have been
periods where ice lobes were polythermal (Darvill et al., 2016).   However, recent ice flow
modelling (Leger et al., 2021b) suggests that varying ice viscosity mainly impacts the
accumulation zone thickness in simulations of paleoglaciers in Northeastern Patagonia, with
minimal impacts on overall glacier length and extent.  Accordingly, based on sensitivity tests (see
supplement section S1), our model is 2-dimensional and we do not solve for ice temperature and
viscosity allowing for increased computational efficiency.  For our purposes, we use Glen's flow
law (Glen, 1955) and set the ice viscosity following the rate factors in Cuffey and Paterson (2010)
assuming an ice temperature of -0.2°C.  We use a linear friction law (Budd et al., 1979)
$$\tau_b = -k^2 N \mathrm{u}_b \tag{1}$$
where  $\tau_b$ represents the basal stress, N represents the effective pressure, and $\mathrm{u}_b$ is the magnitude
of the basal velocity.  Here N = g($\rho_i$H + $\rho_w Z_b$), where g is gravity, H is ice thickness, $\rho_I$ is the
density of ice, $\rho_w$ is the density of water, and $Z_b$ is bedrock elevation following Cuffey and Paterson
206   (2010).

The spatially varying friction coefficient, $k$, is constructed following Åkesson et al. (2018):
$$k = 200 \times \frac{\min[\max(0, z_b + 600), z_b]}{\max(z_b)} \tag{2}$$
where $z_b$ is the height of the bedrock with respect to sea level.  Using this parameterization, basal
friction is larger across high topographic relief and lower across valleys, and areas below sea level.
To account for the influence of glacial isostatic adjustment (GIA), we prescribe a transiently
evolving reconstruction of relative sea level from the global GIA model of the last glacial cycle
from Caron et al. (2018).  This includes three physical components:  1) Bedrock vertical motion
2.) Eustatic sea level, and 3.) Geoid changes.  The time series we use to prescribe GIA is from the
model average of an ensemble of GIA forward model estimations from Caron et al., 2018.  The
prescribed GIA is in good agreement (Figure S2) with a reconstruction of relative sea-level change
from an isolation basin in central Patagonia (Troch et al., 2022).  This methodology has been
applied in recent modelling following Cuzzone et al. (2019) and Briner et al. (2020).
**2.2 Experimental Design**
In order to simulate the ice history at the LGM and across the last deglaciation we use climate
model output from the National Center for Atmospheric Research Community Climate System
Model (CCSM3) TraCE-21ka transient climate simulation of the last deglaciation (Liu et al., 2009;
He et al., 2013).  Monthly mean output of temperature and precipitation are used from these
simulations as inputs to our glaciological model (full climate forcings details are further described
in section 2.4) and we use the monthly mean output every 50 years across the last deglaciation.
Large, multi-proxy reconstructions from He and Clark (2022), Liu et al. (2009), He et al. (2011),
and Shakun et al. (2012; 2015) have all demonstrated good agreement between TRACE 21k and
a wide variety of paleo-proxy data during the last deglaciation that include records from the West
Antarctic and South America.
**2.3 Surface Mass Balance**
In order to simulate the deglaciation of the PIS across our model domain we require inputs of
temperature and precipitation to estimate the surface mass balance.  To derive snow and ice melt
we use a positive degree day model (Tarasov and Peltier, 1999; Le Morzadec et al., 2015; Cuzzone
et al., 2019; Briner et al., 2020).  Our degree day factor for snow melt is 3 mm °C$^{-1}$day$^{-1}$ and 6 mm
°C$^{-1}$day$^{-1}$ for bare ice melt, and we use a lapse rate of 6 °C/km to adjust the temperature of the

climate forcings to surface elevation, which are within a range of typical values used to model contemporary and paleo glaciers across Patagonia (see Fernandez et al., 2016 Table 3; Yan et al., 2022). The hourly temperatures are assumed to have a normal distribution, of standard deviation 3.5 degrees Celsius around the monthly mean. An elevation-dependent desertification is included (Budd and Smith, 1981) which reduces precipitation by a factor of 2 for every kilometer change in ice sheet surface elevation. We note that the values in the surface mass balance parameters were chosen to provide a reasonable fit within 5% between the simulated LGM ice sheet area and the reconstructed ice area from PATICE (see Figure 4 and 10).

## 2.4 Climate forcings

In order to scale monthly temperature and precipitation across the LGM and last deglaciation we applied a commonly used modeling approach (Pollard et al., 2012; Seguinot et al., 2016; Golledge et al., 2017; Tigchlaar et al., 2019; Clark et al., 2020; Briner et al., 2020; Cuzzone et al., 2022; Yan

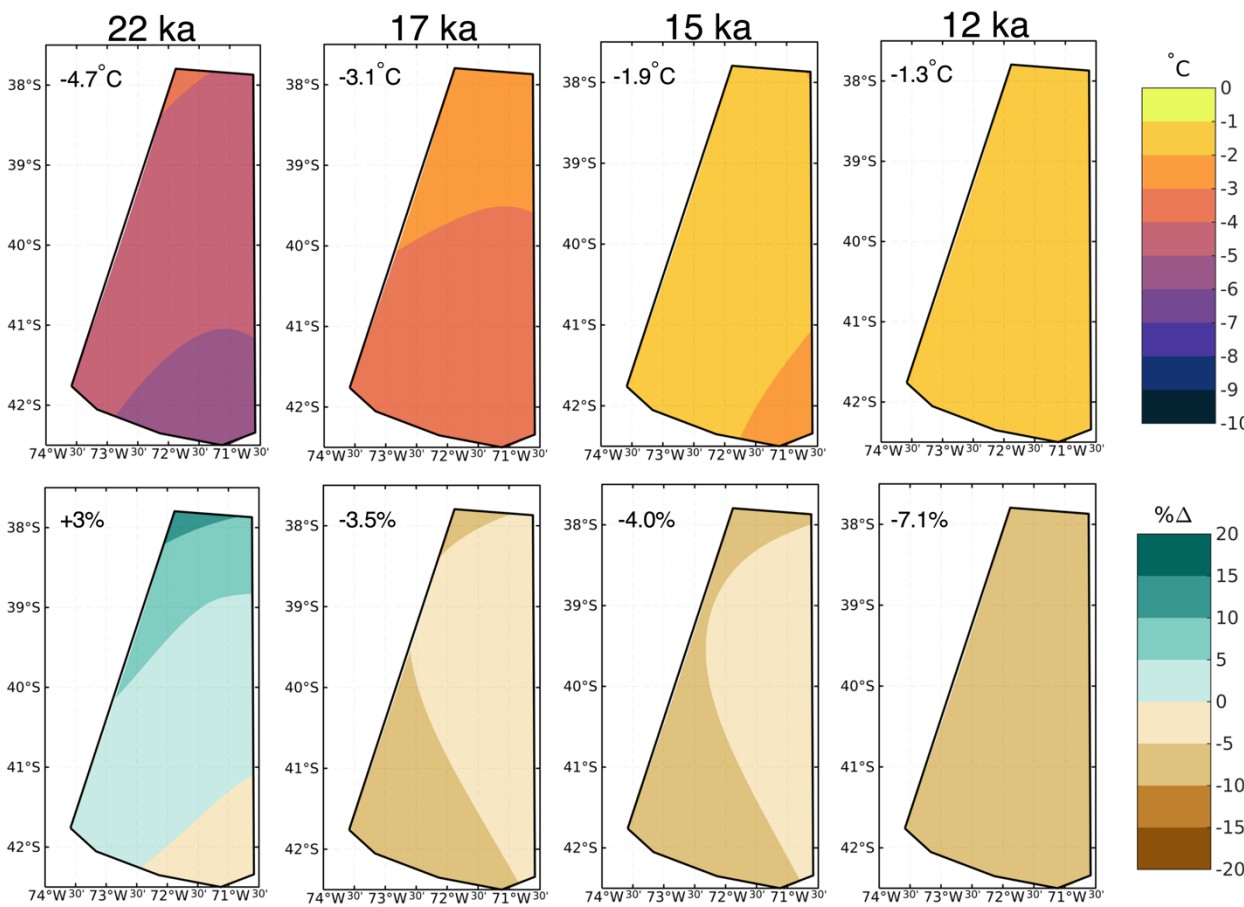

Figure 3. The bilinearly summer (DJF) temperature (top row) and winter (JJA) precipitation anomalies (bottom row) from TraCE-21ka at 22 ka, 17 ka, 16 ka, and 12 ka. Anomalies are taken as the difference between the corresponding time period and preindustrial (LGM-PI), with the precipitation anomalies expressed as the percent difference from preindustrial. The area averaged value of the anomaly is shown in the upper left corner of each

et al., 2022; equations 3 and 4). First, we use the monthly mean climatology of temperature and precipitation for the period 1979-2018 ($\bar{T}_{(1979-2018)}$, $\bar{P}_{(1979-2018)}$) from the Center for Climate

Resilience Research Meteorological dataset version 2.0 (CR2MET; Boisier et al., 2018). This output, which uses information from a climate reanalysis and is calibrated against rain-gauge observations, is provided at 5 km spatial resolution.

We then bilinearly interpolate these fields onto our model mesh.

$$T_t = \bar{T}_{(1979-2018)} + \Delta T_t \tag{3}$$

$$P_t = \bar{P}_{(1979-2018)} + \Delta P_t \tag{4}$$

Next, anomalies of the monthly temperature and precipitation fields from TraCE-21ka (Liu et al., 2009; He et al., 2013) are computed as the difference from the preindustrial control run and interpolated onto our model mesh ($\Delta T_t$ and $\Delta P_t$). These anomalies are added to the contemporary monthly mean as shown in equations 3 and 4, to produce the monthly temperature and precipitation fields at LGM and across the last deglaciation ($T_t$ and $P_t$). In Figure 3 anomalies from preindustrial of summer temperature and winter precipitation are shown for 22 ka, 17 ka, 15 ka, and 12 ka.

**2.5 Ice front migration and iceberg calving**

We simulate calving where the PIS interacts with ocean, but do not include any treatment of calving in proglacial lakes (see section 4.3). We track the motion of the ice front using the level-set method described in Bondzio et al. (2016; equation 3) in which the ice velocity $v_f$, is a function of the ice velocity vector at the ice front (v), the calving rate (c), the melting rate at the calving front ($\dot{M}$), and where n is the unit normal vector pointing horizontally outward from the calving front. For these simulations the melting rate is assumed to be negligible compared to the calving rate, so $\dot{M}$ is set to 0.

$$v_f = v - (c + \dot{M})\, n \tag{5}$$

To simulate calving we employ the more physically based Von Mises stress calving approach (Morlighem et al., 2016) which relates the calving rate (c) to the tensile stresses simulated within the ice, where $\tilde{\sigma}$ is the von Mises tensile strength, $\|v\|$ is the magnitude of the horizontal ice velocity, and $\sigma_{max}$ is the maximum stress threshold which has separate values for tidewater and floating ice, namely 1 MPa and 200 kPa.

$$c = \|v\| \frac{\tilde{\sigma}}{\sigma_{max}} \tag{6}$$

The ice front will retreat if von Mises tensile strength exceeds the user defined stress threshold. This calving law has been applied in Greenland to assess marine terminating icefront stability (Bondzio et al., 2016; Morlighem et al., 2016; Choi et al., 2021; Cuzzone et al., 2022) and for our simulations applies where ocean is present such as the Seno de Reloncaví and the Golfo de Ancud (see Figure 2).

**3 Results**

### 3.1 Simulated LGM state

In order to arrive at a steady state LGM ice geometry, we first initialize our model with an ice-free configuration. A constant LGM monthly climatology of temperature and precipitation are then applied, as well as the prescribed GIA from Caron et al. (2018). We allow the ice sheet to relax for 10,000 years, during which, the ice sheet is free to grow and expand until it reaches a steady state ice geometry and volume, in equilibrium with the climate forcings.

At 22 ka, Trace-21ka simulates an area averaged summertime (DJF) cooling of 4.7°C relative to the PI across our model domain (Figure 3). The LGM cooling increases from north to south, with the greatest magnitude of cooling occurring across the southern portion of our model domain of up to 6°C. During winter (JJA), Trace-21ka simulates an overall wetter climate across our model domain during the LGM relative to the PI. While the area-averaged LGM precipitation anomaly is small (3% higher), the LGM precipitation anomaly increases from south to north, with Trace-21ka simulating 10-15% more wintertime precipitation during the LGM than the PI across the northern portion of the model domain.

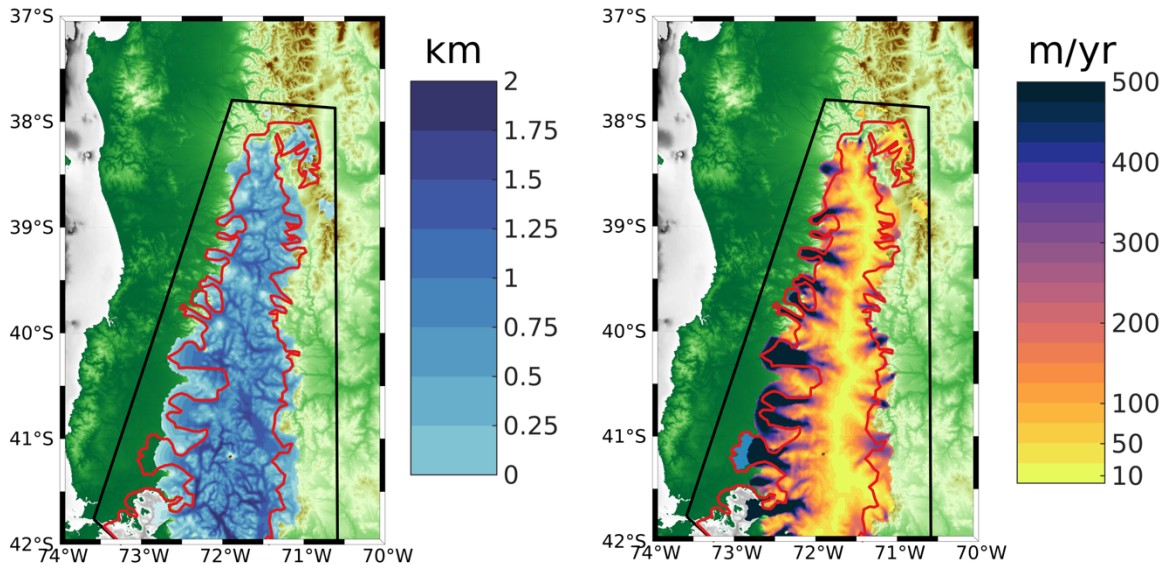

Figure 4. The simulated LGM ice thickness (km; left panel) and the simulated LGM ice surface velocity (km/yr; right panel) is shown. The black outline denotes our ice sheet model boundary, and the red line denotes the LGM reconstructed ice extent from PATICE (Davies et al., 2020).

Bedrock elevation increases from west to east, with deep valleys interspersed across most of our model domain (Figure 2). LGM ice thickness is greatest in these valleys (upwards of 2000 meters) where driving stresses dominate and where bedrock geometry controls the flow of ice from higher terrain and through these valleys (Figure 4). Across the highest terrain such as the many volcanoes across the CLD, ice is comparatively thinner than the surrounding valleys. An ice divide is present as slow ice velocities in the interior of the ice sheet, which give way to fast flowing outlet glaciers especially on the western margin of the CLD where velocities reach in excess of 500 m/yr and in some location up to 2 km/yr. The simulated LGM ice sheet area across the CLD is 414,120 km$^2$,

which is within 1% of the area calculated from the PATICE reconstruction (414,690 km$^2$; Figure
10).  This agreement is in part due to the tuning of our degree day factors as discussed in section
2.3, and gives confidence to our ability to simulate a reasonable LGM ice sheet across the CLD
and throughout the last deglaciation.

**3.2 Simulation of the Last Deglaciation**

Monthly mean temperature and precipitation, taken every 50 years from the TraCE-21ka (Liu et
al., 2009; He et al., 2013) experiment is used to drive our simulation of ice history across the last
deglaciation (22 ka – 10 ka).  The transient simulation is initialized with the LGM ice sheet
geometry shown in Figure 4, and is run forward with the appropriate climate boundary conditions
until 10 ka.

*3.2.1 Pattern of Deglaciation*

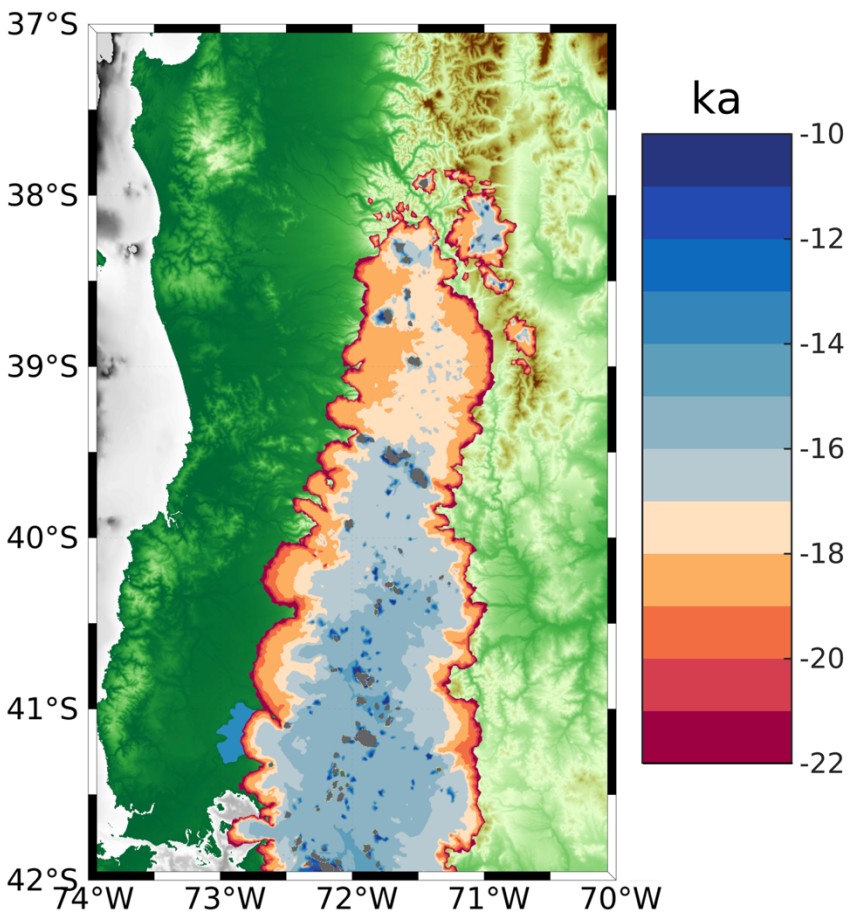

Figure 5.    The simulated deglaciation age for the transient simulation from the LGM to 10 ka.   The gray color indicates where ice persists after 10 ka.

From the resulting transient simulation, we calculate the timing of deglaciation across our model
domain (Figure 5) as the youngest age at which grid points become ice free.  Our map of the
simulated deglaciation can be paired with a timeseries of the rate of ice mass change (Figure 6) to
highlight some key features in the magnitude and timing of ice retreat between 22 ka and 10 ka.
Between 22 ka to 19 ka, the ice sheet undergoes periods of minor to moderate ice mass loss and
gain in an interval of time where summer temperature anomalies (Figure 6) and the corresponding
ice margin remain relatively stable (Figure 5).  Between 19 ka and 18.5 ka, coincident with a rise
in summertime temperature (Figure 6), a pulse of ice mass loss exceeding 5,000 GT/century occurs
before trending toward minimal ice mass loss around 18 ka as the rise in summer temperature
levels off.  During this time interval, the ice margin pulls back considerably towards higher terrain
across the northern portion of the model domain (Figure 5), and many of the fast-flowing outlet
glaciers on the western margin retreat back towards the ice sheet interior.  Between 18 ka to 16.2
ka, summer temperature rises steadily ~1.2°C and is punctuated with an abrupt warming of ~0.5°C
at 16 ka (Figure 6).  During this interval, ice mass loss remains high and steady at ~1000
GT/century with pulses of increased mass loss at 17.8 ka, 16.8 ka, and 16 ka varying between
2000-5000 GT/century (Figure 6).

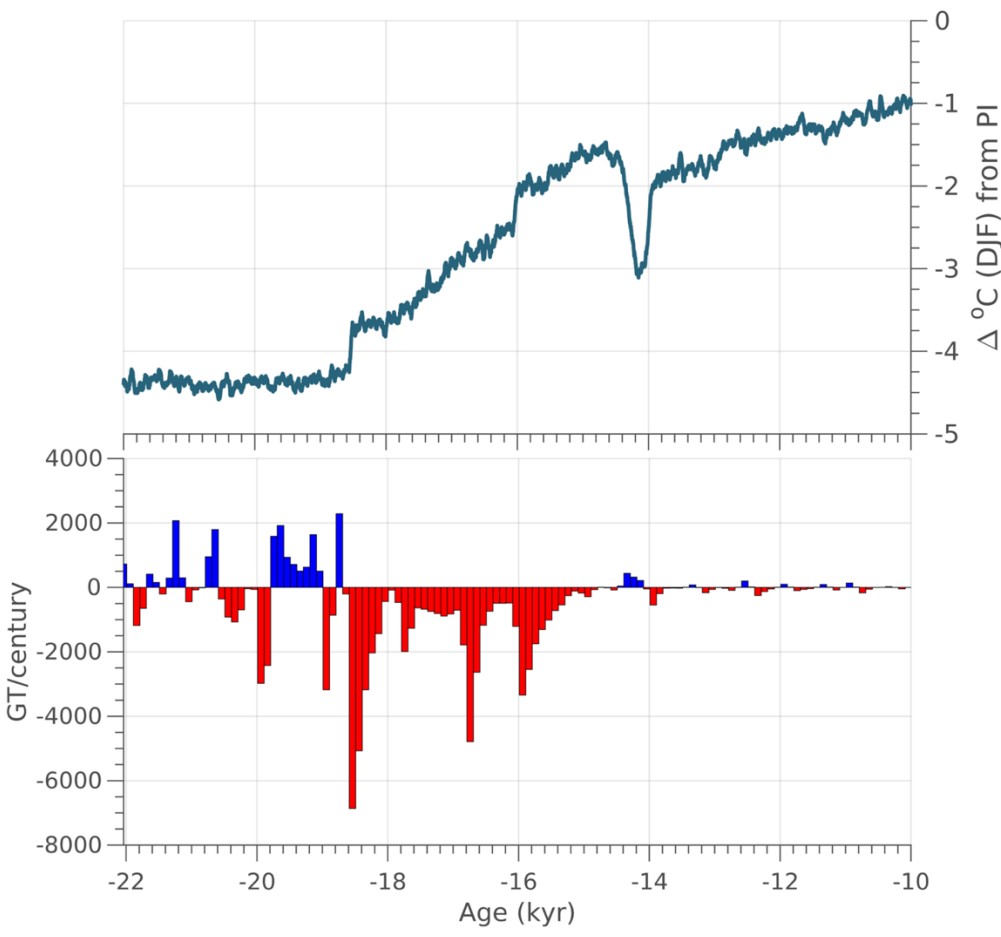

Figure 6.  Top Panel:  The TraCE-21ka Summer (DJF) temperature anomaly taken as the difference from the
preindustrial period, area averaged across our model domain.  Bottom Panel:  The simulated ice mass change
calculated in GT/century across the last deglaciation (22 ka to 10 ka).  Red indicates ice mass loss, and blue indicates
ice mass gain.


By 17 ka, the northern portion of the model domain (north of 39.5°S), has generally become ice
free for the exception of the highest terrain (e.g., mountain glaciers). By 16 ka, between 39.5°S
and 40.5°S, ice remains only on the highest terrain (Figure 5), however ice cover persists south of
40.5°S. Between 16 ka and 15 ka, summer temperature rises ~0.5°C (Figure 6) and the remaining
ice sheet retreats south of 40.5°S. By 15 ka, there is no evidence of an ice sheet, with only
mountain glaciers and small ice caps (e.g., Cerro Tronador) existing across the high terrain
throughout the model domain (Figure 5).
After 15 ka, TraCE-21ka simulates a short and abrupt Antarctic Cold Reversal (ACR) between
14.6 ka and 14 ka (Figure 6), before temperatures continue to rise into the early Holocene. There
is only a minor ice mass gain (e.g., <500 GT/yr) during the ACR, and minimal fluctuation in ice
mass after 14 ka. By 10 ka, only small mountain glaciers persist across the high terrain and
volcanoes of the CLD (gray color in Figure 5).
*3.2.2 Sensitivity Tests*
To better assess how changes in precipitation may modulate the deglaciation across the CLD we
perform additional sensitivity tests. We refer to the simulation discussed above as our *main*
*simulation*, where the climate boundary conditions of temperature and precipitation varied
temporally and spatially across the last deglaciation. Three more simulations are performed where
temperature is allowed to vary across the last deglaciation, but precipitation remains fixed at a
given magnitude for a particular time interval. Each experiment is listed below as:

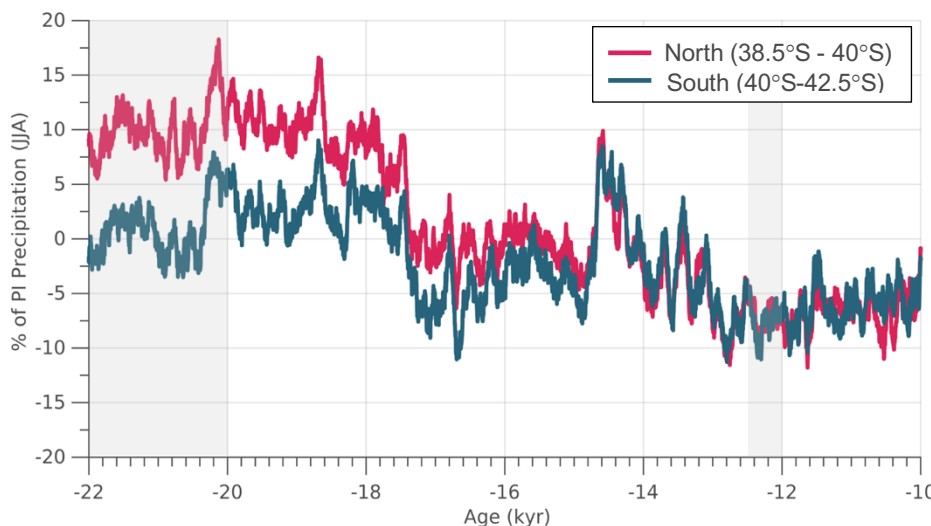

Figure 7. The winter (JJA) precipitation anomaly expressed as the percent difference from the preindustrial period. The area averaged anomaly is shown for the region north of 40°S and for the region south of 40°S (see Figure 2 for reference to the latitudinal range of our model domain). Intervals of time used in the sensitivity tests are highlighted by the gray shading.

*Precip. PI*: Monthly precipitation is held constant at the preindustrial mean. Preindustrial
precipitation is reduced compared to the period 22 ka to18 ka, but is similar to and higher than
what is simulated after 18 ka for the exception of the ACR at 14.5 ka (Figure 7).
*Precip. 12 ka:* Monthly precipitation is held constant at the 12.5 ka-12 ka mean. This is a
period of reduced precipitation relative to the preindustrial (~7% reduction; Figure 7).
*Precip LGM*: Monthly precipitation is held constant to the 22-20 ka mean, which is
approximately 10% higher than preindustrial values across the Northern portion of the model
domain (North of 40°S).
Across our model domain during experiment *Precip. PI* (Figure 8A), wintertime precipitation
during the preindustrial is reduced compared to the early deglaciation (22 ka to 18ka) and is similar
to slightly higher particularly south of 40°S after 18 ka (Figure 7). When holding precipitation
constant at the preindustrial mean through the last deglaciation, the ice retreats faster across most
portions of the model domain, particularly along the ice margins and in area north of 40°S. In the
southern portion of our model domain (south of 40°S), where the changes in deglacial precipitation
relative to the preindustrial are lower (Figure 3 and 7), the difference in simulated deglaciation age
are also smaller. In general, the pace of deglaciation increases by up to 1 kyr compared to the
main simulation, with many locations experiencing deglaciation 200-600 yrs earlier than the main
simulation.

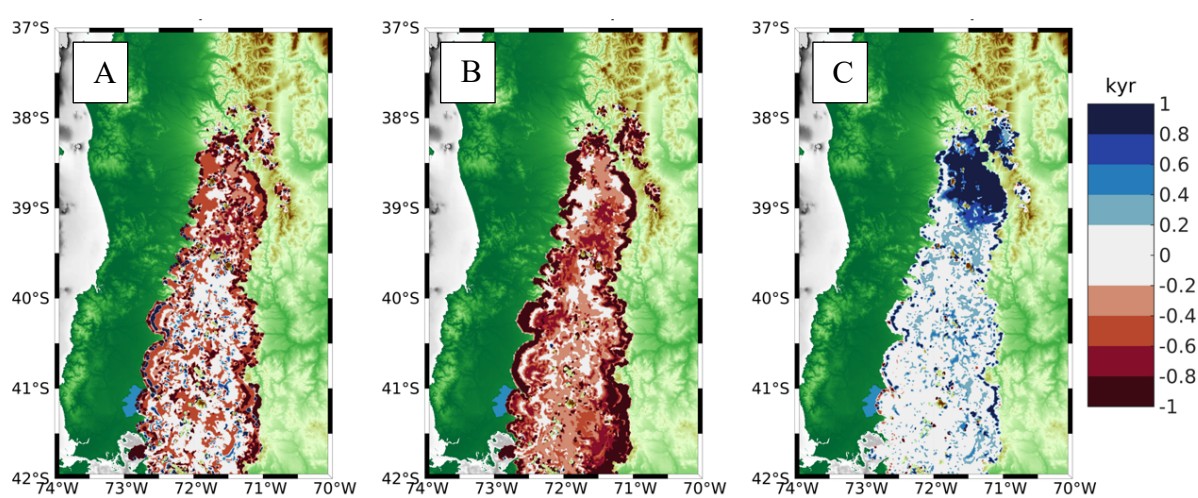

Figure 8. A) The difference in the simulated deglaciation age between sensitivity experiment *Precip. PI:* B.) experiment *Precip. 12 ka* , C.) and experiment *Precip LGM* , from the main simulation. Blue colors indicate slower ice retreat for the sensitivity experiments compared to the main simulation, while red colors indicate faster ice retreat for the sensitivity experiments compared to the main run.

For experiment *Precip. 12 ka*, winter precipitation is reduced by up to 7% (Figure 8B) relative to
the preindustrial across the model domain (Figure 3 and 7). In this experiment ice retreats faster
across most of the CLD, from the ice margins and through the interior. Deglaciation along the
margins occurs >1 kyr faster in many locations, and between 200 yrs to 1 kyr faster across portions
of the ice interior. For experiment *Precip LGM*, winter precipitation is increased by up to 10%
(Figure 8C; *Precip LGM*:) across the northern portion of the model domain (north of 40°S) relative
to preindustrial, but is similar to preindustrial values across the southern portion of our model
domain (south of 40°S). In this experiment, with the imposed higher precipitation across the
northern portion of the model domain, ice retreats slower during the last deglaciation relative to
our standard simulation by >1 kyr, and in some locations up to 2 kyr.

**3.3 Comparison to the reconstructed deglacial ice extent**

Shown in Figure 1, PATICE assigns high to medium confidence to the reconstructed LGM (25 ka
– 20 ka) ice extent along most of the western ice margin and portions of the eastern margin, with
low confidence assigned to the northernmost ice extent.  The majority of the ice history is poorly
constrained (low confidence) during the deglaciation, and PATICE reconstructs a small cap that

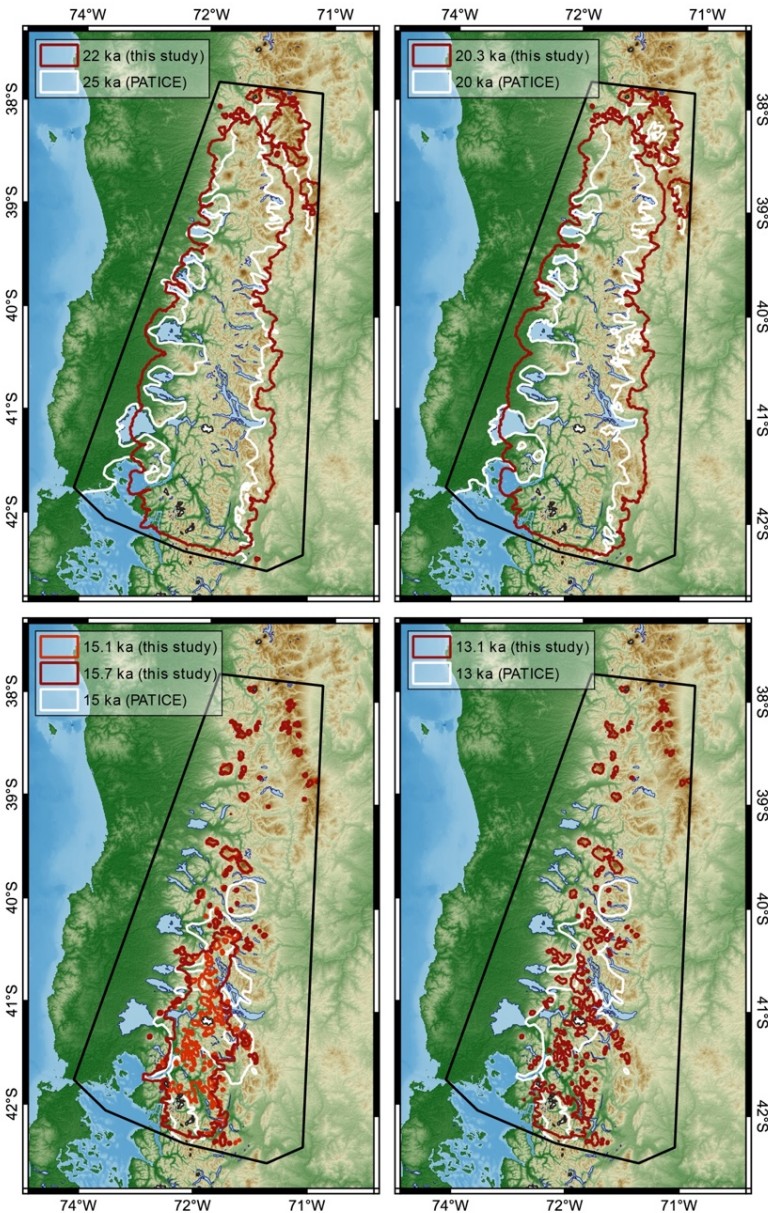

Figure 9.  Comparison between the simulated ice extent at time intervals closest to the corresponding reconstructed ice extent from PATICE (Davies et al., 2020).

persists across the southern CLD until 10 ka, after which the ice disappears and only the Cerro
Tronador glacier remains (see Figure 13 from Davies et al., 2020). We show the simulated and
reconstructed ice extent in Figure 9 as well as the calculated ice area from PATICE at 20 ka, 15
ka, 13 ka, and 10ka and for our transient simulation in Figure 10.  At 22 ka (Figure 9), our model
simulates a generally greater ice extent along the eastern and western margin, except at the Seno
de Reloncaví, Golfo de Ancud, and Lago Llanquihue, where the simulated ice margin does not
advance to the well dated terminal LGM moraines (Mercer, 1972; Porter, 1981; Andersen et al.,
1999; Denton et al., 1999).  At 20 ka, the simulated ice area is $4.1 \times 10^4$ km$^2$ which is nearly identical
to the PATICE areal extent across our model domain (Figure 10).  The ice margin at the Seno de
Reloncaví, Lago Llanquihue, and other locations along the eastern boundary in the CLD advances
slightly at 20 ka, but still remain inboard of the PATICE reconstruction for these regions.
Between 18.3 ka and 15 ka large scale ice retreat occurs, and the simulated ice sheet loses 90% of
its ice area, while the PATICE reconstruction suggests a reduction of 75% (Figure 10).  At 15 ka,
PATICE reconstructs an existing ice cap that separates from the remainder of the PIS to the south
(Figure 9).  This is in contrast to the simulated ice extent, which shows that by 15 ka, the PIS
across our model domain has completely retreated and only mountain glaciers or small ice caps
exist amongst the high terrain.  However, if we compare the PATICE area at 15 ka and the
simulated ice area at 15.7 ka (Figure 10; green rectangle), they are nearly identical at $1.2 \times 10^4$ km$^2$.
While the PATICE ice extent at 15 ka and the simulated ice extent 15.7 ka do not match
completely, the simulated ice extent at 15.7 ka still has evidence of a large ice cap similar to the
PATICE reconstruction.  Therefore, the simulated transition from ice sheet to ice cap and to
discrete mountain glaciers occurs between 15.7 ka and 15 ka in our simulations.  By 13 ka, our
simulated ice area is 60% lower than the PATICE reconstructed area.  By 10 ka this difference is
50%, however by this time the majority of the ice sheet has deglaciated (Figure 10), with our model
simulating discrete mountain glaciers while PATICE reconstructs a small and narrow ice cap
across the high terrain in the southern CLD (also see Figure 1).


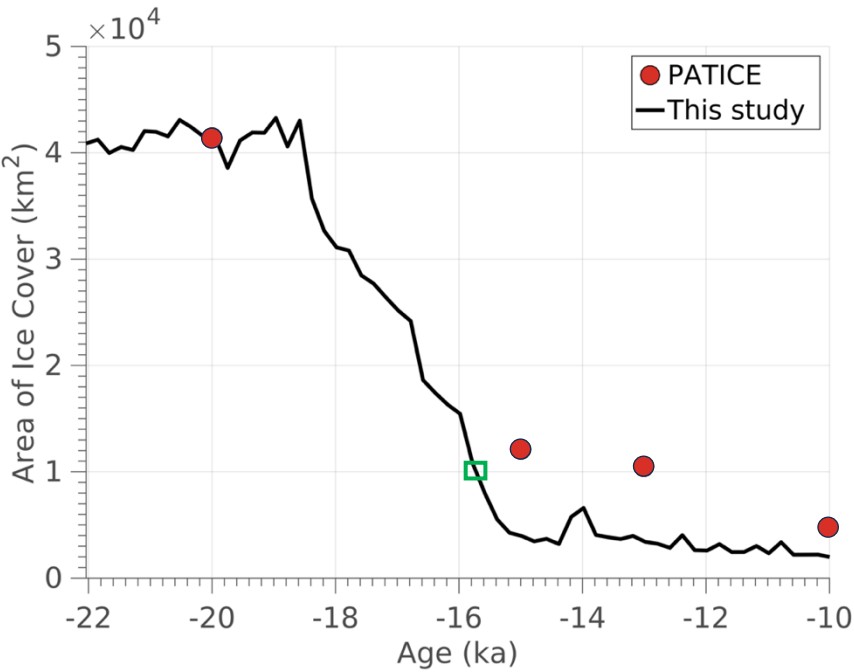

Figure 10. The simulated ice area (km²) from 22 ka to 10 ka shown as the black line. The red dots indicate the calculated ice area across our model domain for the reconstructed ice extent from PATICE (Davies et al., 2020). The green rectangle highlights the simulated ice area at 15.7 ka.


**4 Discussion**

*4.1 Climate-ice sensitivity*

Determining the influence of the SWW on the heat and hydrologic budget across South America
during the LGM and last deglaciation remains difficult, as paleo-proxy data is limited and climate
models tend to disagree on the evolution of the SWW (Kohfeld, 2013; Berman et al., 2018). And
while paleo-proxy evidence does suggest wetter conditions across the CLD during the late glacial
(Moreno and Videla, 2018), linking this variability to changes in the position and strength of the
SWW remains difficult (Kohfeld et al., 2013).

The scale at which we deduce ice history and climate interactions is also important. Looking at
the PIS as a whole, recent numerical ice sheet modelling studies indicate that the simulated ice
extent and volume for the entire PIS at the LGM is largely controlled by the magnitude of the
temperature anomaly compared to present day (Yan et al., 2022). However, regional scale ice
flow modelling informed by geologic constraints on past ice margin extent show that higher
precipitation during the LGM (Leger et al., 2021b), the late glacial, and the Holocene (Muir et al.,
2023; Martin et al., 2022) is needed to support model-data agreement. It appears that during the
LGM a northward shift in the SWW (Kohfeld et al., 2013; Rojas et al., 2009; Togweillier et al.,
2006) or a strengthening or expansion of the wind belt (Lamy et al., 2010) is perhaps the most
likely scenario, with high frequency variability possible during the deglaciation as atmospheric
reorganization altered the heat and hydrologic budget as recorded by glacier and ice sheet change
(Davies et al., 2020; Boex et al., 2013).
We analyzed outputs of the wintertime (JJA) 925 hPa zonal wind as the mean over 500 yr periods
from TraCE-21ka for the LGM (22-21ka), 18ka (18.5-18ka), 16ka (16.5-16ka), 14ka (14.5-14ka),
12ka (12.5-12ka) and the Preindustrial (Supplemental section 3, Figures S3 A-E).  Across our
model domain and to its south, relative to the PI, zonal winds are stronger during the LGM with a
southerly displacement (Figure S3A first and second column). During 18ka (Figure S3B), the zonal
wind increases in strength relative to the PI, with the stronger winds having wider latitudinal
coverage, particularly across our model domain.  While the mean position of the SWW is poleward
at 18ka relative to the PI (Jiang and Yan, 2022), across Patagonia the simulated position of the
maximum zonal wind is at the same latitudinal band as the PI.  At 16ka, the zonal wind is stronger
across our domain and Patagonia (Figure S3C) relative to the PI, although not as large as the
differences during 18ka.  By 14ka, the strength in the zonal winds across Patagonia and our model
domain are similar to slightly stronger than the PI (Figure S3D), however, the zonal wind
maximum is situated more equatorward across our model domain relative to the PI.   By 12ka
(Figure S3E), the zonal wind is similar to slightly weaker than the PI across our model domain,
although it is stronger relative to the PI to the south of our model domain across central and
southern Patagonia.  The position of the maximum zonal winds is also displaced further south
relative to the PI.  These changes in strength and position of the simulated SWW during the last
deglaciation are similar to the findings of Jian and Yan (2020), which found that relative to the
Preindustrial (PI), TraCE-21ka simulates a more poleward subtropical and subpolar jet over the
Southern hemisphere at the LGM.  During the remainder of the LGM and last deglaciation, the
overall position of the SWW migrates northward in TraCE-21ka, with poleward displacements
during Heinrich Stadial 1 (HS1), equatorward displacements during the Antarctic Cold Reversal
(ACR), and poleward displacements during the Younger Dryas (YD), similar to our analysis.
Additionally, we evaluated the wintertime (JJA) low-level (850 hPa) moisture flux convergence
from TraCE-21ka (MFC; Supplement section 4, Figure S4A-E), which is influenced by the mean
flow and transient eddies in the extratropical hydrologic cycle (Peixoto and Oort, 1992).  During
the LGM and 18 ka, MFC increases across our model domain, consistent with a convergence of
the mean flow moisture fields relative to the PI (Figure S4 A, B).  During the LGM and 18ka, we
note that TraCE-21ka simulates higher JJA precipitation anomalies (relative to the PI) across our
model domain (Figure 7).  While our analysis cannot directly constrain the source of the positive
precipitation anomalies (e.g., mean flow, storms), the strength of the simulated SWW in TraCE-
21ka increases across our model domain (Figure S3 A, B) coincident with the increases in MFC,
which may contribute to the positive precipitation anomalies at these time intervals (Figure 7).  By
16ka, there is increased divergence in the 925 hPa winds and moisture relative to the PI (Figure
S4 C).  Decreased MFC relative to the PI coincides with a reduction in precipitation across our
model domain that is similar to or less than the PI (Figure 7).  We note that the ice thickness
boundary conditions used in the TraCE-21ka come from the Ice5G reconstruction (Peltier, 2004),
which has the PIS being completely deglaciated by 16ka.   However, our analysis cannot
decompose whether the simulated changes in precipitation and MFC are a consequence of the
coupling between regional atmospheric circulation and the ice thickness boundary conditions used
in TraCE-21ka or if these changes represent wider interactions with changes in hemispheric
atmospheric circulation.  By 14ka, and during the ACR, MFC increases relative to the PI (Figure
S4D).  This is consistent with a simulated equatorward migration of the SWW as shown in Jiang
and Yan (2020) and our analysis (Figure S3D), and positive anomalies in precipitation across our
model domain relative to the PI (Figure 7).  By 12ka, precipitation across our model domain is
reduced relative to the PI (Figure 3 and 7), and TraCE-21ka simulates a reduction in the MFC as
well as a poleward migration of the SWW (Figure S3E; Jiang and Yan, 2020).
When considering proxy records of precipitation across the CLD, there is reasonable agreement
with the changes in precipitation simulated by TraCE-21ka.   Moreno et al. (1999; 2015) and
Moreno and Videla (2018) find that wetter than present day conditions existed across the CLD
during the LGM and early deglaciation which is consistent with the precipitation anomalies
simulated by TraCE-21ka (Figure 3 and 7).  These changes in paleoclimate proxies are attributed
to an intensified storm track associated with an equatorward shift of the SWW (Moreno et al. 1999;
2015).   While TraCE-21ka instead simulates a poleward shift of the SWW during these time
intervals, increases in precipitation and the intensification of the storm track as inferred by Moreno
et al. (2015) may also be consistent with a strengthening of the SWW as simulated by TraCE-21ka
during these intervals (Figure S3 A, B; Rojas et al., 2009; Sime et al., 2013; Kohfeld et al., 2013).
Moreno et al. (2015) note that rapid warming ensues across the CLD around 17,800 cal yr BP,
which is similar to the timing of deglacial warming as simulated by TraCE-21ka around 18.5 ka
(Figure 6).   Coincident with this rapid temperature rise, Moreno et al. (2015) note a shift from
hyper humid to humid conditions which aligns well with decreases in the simulated precipitation
in TraCE-21ka across our model domain (Figure 7).   Lastly, Moreno et al. (1999; 2015) find that
colder and wetter conditions occur across the CLD during the ACR, and infer an equatorward
expansion of the SWW as a potential cause.   While TraCE-21ka simulates an abrupt and short
ACR, it does simulate an equatorward expansion of the SWW (Figure S4 D; Jian and Yan, 2020),
associated cooling (Figure 6), and increases in precipitation (Figure 7) that agree with the proxy
data.
Prior numerical ice flow modelling has indicated that precipitation played an important role in
controlling the extent of paleoglaciers across the PIS (Muir et al., 2023; Leger et al., 2021b) by
modulating the pace and magnitude of ice retreat and advance during deglaciation (Martin et al.,
2022).  Much of the TraCE-21ka simulated winter precipitation anomalies shown in Figure 3 and
7 are within 10% of the preindustrial value. The sensitivity tests conducted here suggest that
modest changes (~10%) in precipitation can alter the pace of ice retreat across the CLD on
timescales consistent with the resolution of geochronological proxies constraining past ice retreat.
We note that while TraCE-21ka simulates variations in precipitation across our model domain that
are consistent with hydroclimate proxies discussed above (Moreno et al., 1999; 2015; 2018), the
magnitude of those changes is not as large as proxy data across the CLD indicate.  For example,
hydroclimate proxies suggest that the LGM and early deglaciation was up to 2 times wetter across
the CLD than present day (Moreno et al., 1999; Heusser et al., 1999).  Therefore, we can deduce
from our sensitivity analysis here that higher precipitation anomalies during the LGM and last
deglaciation, forced by proposed changes in the SWW (Moreno et al.,1999;2015), may have
helped offset melt from deglacial warming thereby influencing the pacing of early deglacial ice
retreat in this region.
*4.2 Ice retreat during the Last Deglaciation*
The PATICE dataset (Davies et al., 2020) serves as the best available reconstruction of ice margin
change for the PIS across the last deglaciation. This state-of-the-art compilation provides an
empirical reconstruction of the configuration of the PIS as isochrones every 5 ka, from 35 ka to

present, based on detailed geomorphological data and available geochronological evidence. Because geochronological constraints on past PIS change are limited, particularly in the CLD, the PATICE reconstruction assigns qualitative confidence to its reconstructed ice margins. Where there is agreement between geochronological and geomorphological indicators of past ice margin history (i.e., moraines), high confidence is assigned. Where geomorphological evidence suggests the existence of past ice margins, but lacks a geochronological constraint, medium confidence is assigned. Lastly, low confidence is assigned where there is a lack of any indicators of past ice sheet extent, where the ice limits result in interpolated interpretations from immediately adjacent moraines from valleys that have been mapped and dated. Across the CLD, the LGM (25 ka, 20 ka) ice extent is well constrained by geologic proxies particularly in the west and southwest (Figure 1). The moraines that constrain the piedmont ice lobes that formed along the western boundary have reasonable age control (Denton et al., 1999; Moreno et al., 1999; Lowell et al., 1995), giving confidence to the LGM ice margin limits. Beyond this region, age control is sparse along the western boundary for the timing of LGM ice extent, but the existence of well-defined moraines along lakes in the northern CLD are assumed to be in sync with those moraines deposited to the south (Denton et al., 1999). However, low confidence remains in the geologic reconstruction of the LGM ice boundary along the eastern margin where little to no chronological constraints are available. In general, deglaciation from the maximum LGM ice extent begins between 18 – 19 ka (Davies et al., 2020), however, poor age control and a lack of geomorphic indicators make it difficult to constrain the ice extent across this region during the deglaciation. For instance, a single cosmogenic nuclide surface exposure date retrieved from the Nahuel Huapi moraine yielded an age of ~31.4 ka (Zech et al., 2017; 41.04° S, 71.15° W). While it is assumed that the ice limit behaved similarly both to the west and east, the limited existing data prevents a comprehensive understanding of the ice extent at the northeastern margin. This induces the highest level of uncertainty in the reconstruction and hinders our data model comparison. Therefore, we rely on the PATICE dataset interpolated isochrones (low confidence) for this northeastern region as the state-of-the-art reconstruction.

In regards to ice area and extent, our simulated ice sheet at the LGM using TraCE-21ka climate boundary conditions agrees well with the PATICE reconstruction (Figure 10). Our simulations reveal that deglaciation began between 19 ka to 18 ka, consistent with the Davies et al. (2020) reconstruction. Notably, the simulated timing of deglaciation agrees with moraine records further south on the eastern side, such as in Río Corcovado (~43° S, Leger et al., 2021a), Río Cisnes (~44° S, Garcia et al., 2019), Lago Palena/General Vintter (~44° S, Soteres et al., 2022), and Río Ñirehuao (~45° S, Peltier et al., 2023). On the other hand, glaciers are thought to have withdrawn from their LGM position later between ~18 - 17 ka on the northwestern margin (~41° S, Denton et al., 1999; Moreno et al., 2015), in the southern (~46° S, Kaplan et al., 2004), and southernmost regions (~52° S, McCulloch et al., 2000; 2005; Kaplan et al., 2008; Peltier et al., 2021). The simulated ice retreat continues until 15 ka, with the largest pulses in ice mass loss occurring at 18.6 ka, 16.8 ka, and 16 ka (Figure 6). Where PATICE estimates an ice cap around 15 ka (~40°S), our simulations reveal that glaciation was restricted to high elevations. After 15 ka, mountain glaciers remain in our simulation but there is no presence of a large ice cap as reconstructed in PATICE. Comparison between the model simulations and PATICE becomes difficult during the 15 -13 ka period as confidence in the geologic reconstruction is low due to a lack of geochronological and geomorphological constraints on past ice history. Therefore, our model results offer a different reconstruction to PATICE, and indicate that the ice sheet in this region largely retreated by 15 ka,

with only mountain glaciers remaining. This is supported further south, where the ice sheet
disintegrated at ~16 ka with paleolake draining to the Pacific Ocean (~43° S, Leger et al., 2021a)
and the ice remaining limited to higher mountain areas. However, during this interval, the Antarctic
Cold Reversal (ACR) may have influenced the heat and hydrologic budget across this region, with
wetter and cooler conditions interrupting the deglacial warming (Moreno et al., 2018). While
TraCE-21ka simulates a cooler and wetter ACR, it is short-lived, lasting about 500 years as
compared to 2,000 years in some ice core records or proxy-based studies (Lowry et al., 2019; He
et al., 2013, Pedro et al., 2015). This potential for a favorable and prolonged period of glacier
growth is likely missing in our simulations during the ACR.
*4.3 Limitations*
Currently ISSM is undergoing model developments to include a full treatment of solid earth-ice
and sea-level feedbacks (Adhikari et a., 2016). Therefore, at this time, there is no coupling
between the ice sheet and solid earth. Instead, we prescribed GIA from a global GIA model of the
last glacial cycle from Caron et al. (2018). While this model reasonably estimates GIA across the
PIS over the last deglaciation, our simulated ice history does not feedback onto GIA. The ice
history for Patagonia incorporated into the Caron et al. (2018) ensemble is from Ivins et al. 2011.
Therefore, the prescribed GIA response across our domain does not perfectly match our simulated
ice history. Additionally, the global mantle from Caron et al. (2018) does not exhibit regional low
viscosity that is attributable to Patagonia and therefore, current rates of deformation are likely
underestimated by the model. By not simulating the 2-way coupled ice and solid-earth
interactions, we could be missing some feedbacks between our simulated ice history and the solid
earth that may modulate the deglaciation across this region. Despite this limitation however, our
prescribed GIA from Caron et al. (2018) is reasonable when compared with reconstructed deglacial
GIA in Patagonia (Troch et al., 2022; see Figure S2), giving confidence that our simulation is
capturing the regional influence of GIA on the simulated ice history.
Across most of our domain, moraines formed of glacio-tectonized outwash (Bentley, 1996)
provide evidence for an advance of piedmont glaciers across glacial outwash during the LGM,
which formed the physical boundary for some of the existing terminal moraines around the lakes
within the CLD (Bentley, 1996; Bentley, 1997). The formation of ice-contact proglacial lakes
likely occurred as a function of deglacial warming as ice retreated into overdeepings in the
bedrock topography and filled with meltwater (Bentley, 1996). Where there were proglacial lakes
along the westward ice front in the CLD, evidence suggests that ice was grounded during the LGM
(Lago Puyehue; Heirman et al., 2011). During deglaciation, proglacial lakes formed along the ice
sheet margin (Bentley 1996,1997; Davies et al., 2020), with evidence suggesting that local
topography and calving may have influenced the spatially varying retreat rates along these margins
(Bentley, 1997). Recent glacier modelling (Sutherland et al., 2020) suggests that inclusion of ice-
lake interactions may have large impacts on the magnitude and rate of simulated ice front retreat,
as ice-lake interactions promote greater ice velocities, ice flux to the grounding line, and surface
lowering. However, it is not well constrained how the proglacial lakes in the CLD may have
influenced local deglaciation (Heirman et al., 2011). While more geomorphic data is needed,
recent work south of our study region (46.5°S) reconstructed early deglacial ice retreat using a
glaciolacustrine varve record from Lago General Carrera-Buenos Aires (Bendle et al., 2019). The
authors find that following initial retreat due to deglacial warming, the ice margin retreated into a
deepening proglacial lake which accelerated ice retreat in this region due to persistent calving,
therefore supporting the role proglacial lakes likely played across the margins of the retreating PIS
during the last deglaciation. Because the inclusion of ice-lake interactions is relatively novel for
numerical ice flow modeling (Sutherland et al., 2020; Quiquet et al., 2021; Hinck et al., 2022), we
choose to not simulate the evolution and influence of proglacial lakes on the deglaciation across
this model domain. Given this limitation, our simulated magnitude and rate of ice retreat at the
onset of deglaciation may be underestimated, especially when looking at local deglaciation along
these proglacial lakes. Although we do not think that these processes would greatly influence our
conclusions regarding the role of climate on the evolution of the PIS is the CLD and the simulated
ice retreat history, future work is required to assess the influence of proglacial lakes in this region.
**5 Conclusions**
In this study, we use a numerical ice sheet model to simulate the LGM and deglacial ice history
across the northernmost extent of the PIS, the CLD. The ice sheet model used inputs of
temperature and precipitation from the TraCE-21ka climate model simulation covering the last
22,000 years in order to simulate the deglaciation of the PIS across the CLD into the early
Holocene.
Our numerical simulation suggests that large scale ice retreat occurs after 19 ka coincident with
rapid deglacial warming, with the northern portion of the CLD becoming ice free by 17 ka. The
simulated ice retreat agrees well with the most comprehensive geologic assessment of past PIS
history available (PATICE; Davies et al., 2020) for the LGM ice extent and early deglacial but
diverge when considering the ice geometry at and after 15 ka. In our simulations, the PIS persists
until 15 ka across the remainder of the CLD, followed by ice retreat to higher elevations as
mountain glaciers and small ice caps persist into the early Holocene (e.g., Cerro Tronador). The
geologic reconstruction from PATICE instead estimates a small ice cap persisting across the
southern portion of high terrain in the CLD until about 10 ka. However, of the limited geologic
constraints particularly after 15 ka, high uncertainty in the timing and extent of deglacial ice history
remains in the geologic reconstruction. Therefore, our results provide an additional reconstruction
of the deglaciation of the PIS across the CLD that differs from PATICE after 15 ka, emphasizing
a need for future work that aims to improve geologic reconstructions of past ice margin migration
particularly during the later deglaciation across this region.
While deglacial warming was a primary driver of the demise of the PIS across the last deglaciation,
we find that precipitation modulates the pacing and magnitude of deglacial ice retreat across the
CLD. Paleoclimate proxies within the CLD has shown that the strength and position of the SWW
varied during the LGM and last deglaciation, altering hydrologic patterns and influencing the
deglacial mass balance. We find that the simulated changes in the strength and position of the
SWW in TraCE-21ka are similar to those inferred from paleoclimate proxies of precipitation,
consistent with a wetter than preindustrial climate being simulated and reconstructed over the CLD
and in particular the region north of 40°S. Through a series of sensitivity tests, we alter the
magnitude of the precipitation anomaly modestly (up to 10%) during our transient deglacial
simulations and find that the pacing of ice retreat can speed up or slow down by a few hundred
years and up to 2000 years depending on the imposed increase or decrease in the precipitation
anomaly. While paleoclimate proxies of precipitation suggest that the CLD may have experienced

twice as much precipitation during the LGM and early deglacial relative to present day (Moreno et al.,1999;2015), TraCE-21ka simulates smaller increases in LGM and early deglacial precipitation (~10-15% greater than preindustrial). Therefore, while our modelling suggests that modest changes in precipitation can modulate the pace of deglacial ice retreat across the CLD, from our analysis we can deduce that larger anomalies in precipitation as found in the paleoclimate proxies may have an even larger impact on modulating deglacial ice retreat. Because paleoclimate proxies of past precipitation are often lacking, and climate models can simulate a range of possible LGM and deglacial hydrologic states, these results suggest that improved knowledge of the past precipitation is critical towards better understanding the drivers of PIS growth and demise, especially as small variations in precipitation can modulate ice sheet history on scales consistent with geologic proxies.

**Code/Data Availability**

The simulations performed for this paper made use of the open-source Ice-Sheet and Sea-level System Model (ISSM) and are publicly available at https://issm.jpl.nasa.gov/ (Larour et al., 2012).

**Author Contribution**

JC and SM secured funding for this research. JC, MR, and SM all contributed to the project design. JC performed the model setup and simulations. JC performed the analyses on model output, with help from MR who performed analysis on PATICE reconstructions. JC wrote the manuscript with input from MR and SM.

**Competing interests**

The contact author has declared that none of the authors has any competing interests.

**Acknowledgements**

This work was supported by a grant from the National Science Foundation, Frontier Research in Earth Sciences # 2121561. We would like to thank Lambert Caron from the Jet Propulsion Laboratory for his input regarding Glacial Isostatic Adjustment across our study region.

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
