# Peer review of "Modeling the timing of Patagonian Ice Sheet retreat in the Chilean Lake District from"

_The Cryosphere, 2023_

## Referee Comment (RC1)

The authors present a modelling reconstruction of the northern branch of the Patagonian Ice Sheet (PIS) during the last deglaciation, precisely from the last glacial maximum (LGM) to 10 ka ago. The ice-flow model ISSM is used for this purpose. The exercise makes use of various glacial climatologies to reconstruct the glacial state of the ice sheet and of a transient climatology (TraCE-21ka) to simulate the early stage of the deglaciation. The results are then compared with reconstructions available for that region (PATICE - Davies et al., 2020). Ice-climate interactions, sensitivity on the employed climate model, and goodness of the model results are then discussed.

The work is novel and very appealing, as it is the first modelling work trying to delucidate the deglaciation history of a region of the PIS that unfortunately still presents a lot of uncertainty. The manuscript is well written, well organised ad the methodology is mostly sound. Still, I am very surprised that no Glacial Isostatic Adjustment (GIA) is taken into account in their simulations. I understand the authors justify this deficiency as ISSM is currently lacking a GIA solver (even though they claim it's under development), and that they have already successfully applied a similar model configuration - with only relative sea level changes - in previous work in Greenland (Cuzzone et al., 2019 and Briner et al., 2020). However, to me this lack represents a big flaw in the work presented here, as I would expect that GIA has a clear primary control on the evolution of the PIS during the last deglaciation.

It is very well known that the chilean coastal region presents a unique tectonic setting promoting fast response of solid earth to ice mass changes (e.g. Richter et al., 2016, Troch et al., 2022, …). In fact, the thin lithosphere combined with a low upper mantle viscosity and current fast ice retreat leads to extraordinary high uplift rates there. These are found today near the Southern Patagonia Icefield (~4 cm/a) (Lange et al., 2014; Dietrich et al., 2010), for instance, where the current uplift is mostly due to the ice unload after the Little Ice Age. Yet, strong GIA signals are also found for the last deglaciation/early-middle Holocene in:
- Larenas Bay (48°S, between modern Northern and Southern Patagonia Icefields), which rebounded isostatically by almost 100 m between 16-8 ka ago, with a rate of 1.3 cm/yr (Troch et al., 2022), and most of it occurring before 14 ka ago;
- Northern Patagonia Icefield, where GIA rates of 1.5-3.4 cm/a were found for the past ~8 ka (Bourgois et al., 2016);
- More to the south, in the Strait of Magellan, with uplift rates of 0.5 cm/a for the last 13 ka (Rios et al., 2020).

It is true that data of uplift rates for the last deglaciation/early Holocene, especially in northern Chile, is somehow lacking. However, new relative sea level (RSL) reconstructions along the south-central chilean coast (18°S - 44°S, Garrett et al., 2020) reveal interesting informations. Regions where tectonic deformation is associated with the subduction of the Nazca Plate beneath the South American Plate may have experienced uplift rates even higher than 1 m/ka during the mid Holocene (such as Isla Santa María and Isla Mocha). Other regions, such as the southern area of Bío Bío, Valdivia and Arauco, show a sea level highstand of 6-8 m around 8-7 ka ago compared to the present, suggesting a clear local response of solid Earth to the ice unload. This is confirmed by the agreement between such RSL data and GIA model simulations (ICE-5G and ICE-6G), suggesting that the isostatic uplift has a primary control in changes in RSL in these regions (Garrett et al., 2020). This is also in agreement with strong uplift rates (1.5 ± 0.3 m/ka) found in Isla Santa María during MIS3 (Jara-Muñoz and Melnick, 2015), and those found in the region south of the Arauco Peninsula (0.5 m/ka) during the Holocene (Stefer et al. 2010). All these areas are close to the northern branch of the PIS (35°-43°S), therefore it is very likely that the isostatic rebound had a crucial role also in the evolution of the PIS.

The lack of a clear response of the lithosphere to the ice unload during the deglaciation might well affect the results presented in this paper. In fact, the exclusion of an isostatic rebound due to ice melt might partly explain why the area modelled after 16 ka is well below the reconstructions from PATICE: the modelled surface elevation might be too low to sustain the existence of an extensive ice field for increasing Holocene temperature. This is somehow - although indirectly - shown in the sensitivity experiment, where a constant, LGM precipitation (higher than today) is applied to the whole Holocene. In this test, we see that even a small variation in the precipitation might strongly affect the retreat in terms of timing and deglaciation rate. This is because - here - a higher winter accumulation rate helps to sustain the presence of glaciers even with warmer temperatures. Still, a delayed retreat could also be the result of an uplifted topography, which ensures temperature lapse rate near the glacier surfaces to decrease more rapidly. It would be interesting to see if the

authors can reproduce similar results in the retreat either by reducing the tropospheric lapse rate (i.e. making the atmosphere cool more rapidly at increasing elevations) or by considering a synthetic higher topography (+50 m, +100 m, for example) for specific deglaciation times. Therefore, the deglaciation history might not be only defined by the applied paleo climatology, but also by climatology/topography effects that are not taken into account for the moment due to the lack in the GIA treatment. In such a case the whole discussion on climate-ice sensitivity as described in the manuscript risks becoming pointless. Put it in another words, it might be that the simulation experiment with a higher precipitation set for the whole deglaciation matches better the geological reconstructions, but for the wrong reasons.

Finally, the glacial outline from PATICE (Davies et al., 2020) is really uncertain in the northern part of the domain (in fact only very few radiocarbon dates were taken in the Lake district). It might be that the PIS was covering a region further north than what is presented here, as suggested by previous work (Rabassa & Clapperton 1990, Garret et al., 2020). In that case the reconstruction from Trace-21ka would not be sufficient to cover those areas and the whole discussion comparing different climate model outputs and comparing model results to the reconstructed ice retreat during the deglaciation becomes sterile. Therefore, I suggest to clearly discuss the uncertainties in the glacial reconstruction of the northern boundaries before comparing them to the model simulations.

In summary, I am afraid I cannot recommend the publication of this work in The Cryosphere until the lack of a GIA treatment in the model is either exhaustively discussed, or it is taken into account presenting a new set of simulations from the same ISSM model, when the GIA module becomes available, or from another ice-flow model which already computes the interactions between the solid Earth and the ice sheet.

Here I note down specific comments:
Line 71: please describe how the SWW position and strength changed during the last deglaciation with more details.
Line 85: what do you mean by "climate?"
Line 113-115: This is already written some lines above. Please rephrase to avoid repetitions.
Line 145: how is N calculated?
Lines 156-161: this paragraph about the missing GIA model definitely needs further development. See my main comments above.
Lines 228-229: why not simulating calving at the ice-lake interface too? Could you apply the von mises stress law also there?
Line 248: please change "grounded" to "tidewater".
Line 276: MIROC has drier winter conditions only in the southern part of the domain, Please correct.
Line 399-413: I don't see the point of this sensitivity test. Yes, the experiment might be interesting to see the effect of the increased precipitation in the retreat. But what do we learn from this? Does this mean that the reconstructed precipitation is wrong? Or could this be related (also) to the missing uplift upon ice unload in your experiments? Please, discuss this further and think about other possible sensitivity tests about atmospheric lapse rate/synthetic elevation (see paragraph above).
Line 450 onwards: I am missing at least a large paragraph concerning the model limitations, such as lacking GIA effect, lake-terminating calving, …, and their possible influence in the results.
Line 478: why did you choose this "small sample" of PMIP4 climatologies? Why these models, precisely?
Lines 528: again, could not this be related to the missing regional uplift too?
Lines 557-565: I suggest to work on these conclusions as they are only partly corroborated by your sensitivity test.

Figures:
Figure 1, figure 2: it would be helpful to add some reference locations to the map (e.g. gulf of Ancud, Seno de Reloncaví, …) and lat/lon.
Figure 3: I would like to see a further discussion about the reasons that explain the main differences between the PMIP4 models (model parametrisations, …) or at least citing some papers that point to that.

Figure 6: why not plotting the same figure with respect to the glaciated area and comparing it to PATICE reconstruction? It could be also interesting to plot the same figure, but separately for the northern and southern parts of the domain (e.g. north and south of 40°S) since climatologies present a strong latitudinal pattern.

Figure 7: please choose a different color scale as ice lost from the LGM to 17 ka ago is very difficult to see.

Figure 10: please use different colours for the simulated outlines (orange, red?) otherwise they can be confused with the topography.

References:

Bourgois, J., et al., 2016. Geomorphic records along the general Carrera (Chile)-Buenos Aires (Argentina) glacial lake(46°-48°S), climate inferences, and glacial rebound for the past 7-9 ka. J. Geol.124, 27e53. https://doi.org/10.1086/684252.

Briner et al., 2020, Rate of mass loss from the Greenland Ice Sheet will exceed Holocene values this century. 2020. Nature, 6, 70–74, https://doi.org/10.1038/s41586-020-2742-6.

Cuzzone et al., 2019, The impact of model resolution on the simulated Holocene retreat of the southwestern Greenland ice sheet using the Ice Sheet System Model (ISSM). 2019. The Cryosphere, 13, 879–893, https://doi.org/10.5194/tc-13-879-2019.

Davies et al., 2020. The evolution of the Patagonian Ice Sheet from 35 ka to the present day (PATICE). Earth Sci. Rev. 204, 103152. https://doi.org/10.1016/j.earscirev.2020.103152.

Dietrich, R., et al. "Rapid crustal uplift in Patagonia due to enhanced ice loss." Earth and Planetary Science Letters 289.1-2 (2010): 22-29.

Jara-Mu~noz, J., Melnick, D., 2015. Unravelling sea-level variations and tectonic uplift in wave-built marine terraces, Santa María Island, Chile. Quat. Res. 83, 216e228

Lange, H., et al. "Observed crustal uplift near the Southern Patagonian Icefield constrains improved viscoelastic Earth models." Geophysical Research Letters 41.3 (2014): 805-812.

Rabassa, Jorge, and Chalmers M. Clapperton. "Quaternary glaciations of the southern Andes." Quaternary Science Reviews 9.2-3 (1990): 153-174.

Richter, A., et al., 2016. Crustal deformation across the southern patagonian icefield observed by GNSS. Earth Planet Sci. Lett. 452, 206e215. https://doi.org/10.1016/j.epsl.2016.07.042.

Ríos, F., et al., 2020. Environmental and coastline changes controlling Holocene carbon accumulation rates in fjords of the western Strait of Magellan region. Continent. Shelf Res. 199, 104101. https://doi.org/10.1016/j.csr.2020.104101

Stefer, S., et a., 2010. Forearc uplift rates deduced from sediment cores of two coastal lakes in south-central Chile. Tectonophysics 495, 129e143.

Troch, Matthias, et al. "Glacial isostatic adjustment near the center of the former Patagonian Ice Sheet (48° S) during the last 16.5 kyr." Quaternary Science Reviews 277 (2022): 107346.

---

## Author Comment (AC1)

- **Please note that the reviewer comments are posted in Black, responses posted in Red, and our revisions to the text posted in Blue.**

- The authors present a modelling reconstruction of the northern branch of the Patagonian Ice Sheet (PIS) during the last deglaciation, precisely from the last glacial maximum (LGM) to 10 ka ago. The ice-flow model ISSM is used for this purpose. The exercise makes use of various glacial climatologies to reconstruct the glacial state of the ice sheet and of a transient climatology (TraCE-21ka) to simulate the early stage of the deglaciation. The results are then compared with reconstructions available for that region (PATICE - Davies et al., 2020). Ice-climate interactions, sensitivity on the employed climate model, and goodness of the model results are then discussed.
- The work is novel and very appealing, as it is the first modelling work trying to delucidate the deglaciation history of a region of the PIS that unfortunately still presents a lot of uncertainty. The manuscript is well written, well organised ad the methodology is mostly sound.

We would like to thank the reviewer and appreciate the thorough comments, especially those involving the role of GIA during the deglaciation of the PIS. We apologize for any confusion that was presented in our first draft, but as we describe below, we did prescribe GIA in our numerical simulations and account for its influence during the deglaciation.

- Still, I am very surprised that no Glacial Isostatic Adjustment (GIA) is taken into account in their simulations.

Apologies, but it seems we have introduced some confusion in our original draft.

We account for the GIA modeled time series of bedrock and geoid from Caron et al., 2018 (Global GIA model for the last glacial cycle (-122 to PD)). We include 3 physical components: 1) Bedrock vertical motion 2.) Eustatic sea level 3.) Geoid changes. This is represented by the figure below (Figure S1), showing the transiently evolving relative sea-level changes prescribed in our model simulations across the last deglaciation. The prescribed RSL is on par with a study that the reviewer cited below (i.e. Troch et al., 2022) who use isolation basins to reconstruct RSL across the last deglaciation.

The time series we use to prescribe GIA is from the model average of an ensemble of GIA forward model estimations from Caron et al., 2018.

Given this, and to add clarity to our original text, we adjust lines 156-163. We also would be happy to supply the figure below in a supplement to our manuscript:

"To account for the influence of glacial isostatic adjustment (GIA), we prescribe a transiently evolving reconstruction of relative sea level from a global GIA model of the last glacial cycle from Caron et al. (2018). This includes 3 physical components: 1) Bedrock vertical motion 2.) Eustatic sea level, and 3.) Geoid changes. The time series we use to prescribe GIA is from the model average of an ensemble of GIA forward model estimations from Caron et al., 2018. The prescribed GIA is in good agreement (Figure S1) with a reconstruction of relative sea-level change from an isolation basin in central Patagonia (Troch et al., 2022). This methodology has been applied in recent modelling following Cuzzone et al. (2019) and Briner et al. (2020)."

[Figure]

Figure S1: The time dependent prescribed relative sea level change area averaged across our model domain. The relative sea-level change consists of 1) Bedrock vertical motion, 2.) Eustatic sea level change, and 3.) Geoid changes from Caron et al., 2018 across the last deglaciation.

Limitations section:

We acknowledge that there is no 2-way coupling between the ice and solid-earth in our model currently. For these simulations GIA is prescribed following what was stated above. While the simulated ice history in our experiments is influenced by time varying GIA, the simulated ice changes that occur in our model do not feedback onto GIA. The ice history for Patagonia incorporated into the Caron et al. (2018) ensemble is from Ivins et al. (2011). Therefore, the prescribed GIA response across our domain does not perfectly match our simulated ice history. One can also acknowledge that the model of Caron et al. (2018) is not perfect. The global mantle from Caron et al. (2018) does not exhibit regional low viscosity that is attributable to Patagonia (Personal Communication with Lambert Caron). Therefore, current rates of deformation are likely underestimated by the model.

These are limitations of our model when it comes to GIA. However, given the reasonable agreement of the prescribed GIA and what others have found through direct observations (cited in the reviewer comments), we think our model includes an adequate treatment of the influence of GIA on the simulated ice history. We hope that this helps to clarify that we do indeed account for GIA in our model simulations.

To further address the reviewer comment, we have added a limitations section (see Discussion 4.3).

"Currently ISSM is undergoing model developments to include a full treatment of solid earth-ice and sea-level feedbacks (Adhikari et a., 2016). Therefore, at this time, there is no coupling between the ice sheet and solid earth. Instead, we prescribed GIA from a global GIA model of the last glacial cycle

from Caron et al. (2018). While this model reasonably estimates GIA across the PIS over the last deglaciation, our simulated ice history does not feedback onto GIA. The ice history for Patagonia incorporated into the Caron et al. (2018) ensemble is from Ivins et al. (2011). Therefore, the prescribed GIA response across our domain does not perfectly match our simulated ice history. Additionally, the global mantle from Caron et al. (2018) does not exhibit regional low viscosity that is attributable to Patagonia and therefore, current rates of deformation are likely underestimated by the model. By not simulating the 2-way coupled ice and solid-earth interactions, we could be missing some feedbacks between our simulated ice history and the solid earth that may modulate the deglaciation across this region. Despite this limitation however, our prescribed GIA from Caron et al. (2018) is reasonable when compared with reconstructed deglacial GIA in Patagonia (Troch et al., 2022), giving confidence that the simulations are capturing the regional influence of GIA on the deglacial ice history. "

- It is very well known that the chilean coastal region presents a unique tectonic setting promoting fast response of solid earth to ice mass changes (e.g. Richter et al., 2016, Troch et al., 2022, …). In fact, the thin lithosphere combined with a low upper mantle viscosity and current fast ice retreat leads to extraordinary high uplift rates there. These are found today near the Southern Patagonia Icefield (~4 cm/a) (Lange et al., 2014; Dietrich et al., 2010), for instance, where the current uplift is mostly due to the ice unload after the Little Ice Age. Yet, strong GIA signals are also found for the last deglaciation/early-middle Holocene in:
  - Larenas Bay (48°S, between modern Northern and Southern Patagonia Icefields), which rebounded isostatically by almost 100 m between 16-8 ka ago, with a rate of 1.3 cm/yr (Troch et al., 2022), and most of it occurring before 14 ka ago;
  - Northern Patagonia Icefield, where GIA rates of 1.5-3.4 cm/a were found for the past ~8 ka (Bourgois et al., 2016);
  - More to the south, in the Strait of Magellan, with uplift rates of 0.5 cm/a for the last 13 ka (Rios et al., 2020).
- It is true that data of uplift rates for the last deglaciation/early Holocene, especially in northern Chile, is somehow lacking. However, new relative sea level (RSL) reconstructions along the southcentral chilean coast (18°S - 44°S, Garrett et al., 2020) reveal interesting informations. Regions where tectonic deformation is associated with the subduction of the Nazca Plate beneath the South American Plate may have experienced uplift rates even higher than 1 m/ka during the mid Holocene (such as Isla Santa María and Isla Mocha). Other regions, such as the southern area of Bío Bío, Valdivia and Arauco, show a sea level highstand of 6-8 m around 8-7 ka ago compared to the present, suggesting a clear local response of solid Earth to the ice unload. This is confirmed by the agreement between such RSL data and GIA model simulations (ICE-5G and ICE-6G), suggesting that the isostatic uplift has a primary control in changes in RSL in these regions (Garrett et al., 2020). This is also in agreement with strong uplift rates (1.5 ± 0.3 m/ka) found in Isla Santa María during MIS3 (Jara-Muñoz and Melnick, 2015), and those found in the region south of the Arauco Peninsula (0.5 m/ka) during the Holocene (Stefer et al. 2010). All these areas are close to the northern branch of the PIS (35°-43°S), therefore it is very likely that the isostatic rebound had a crucial role also in the evolution of the PIS.

- The lack of a clear response of the lithosphere to the ice unload during the deglaciation might well affect the results presented in this paper. In fact, the exclusion of an isostatic rebound due to ice melt might partly explain why the area modelled after 16 ka is well below the reconstructions from PATICE: the modelled surface elevation might be too low to sustain the existence of an extensive ice field for increasing Holocene temperature. This is somehow - although indirectly - shown in the sensitivity experiment, where a constant, LGM precipitation (higher than today) is applied to the whole Holocene. In this test, we see that even a small variation in the precipitation might strongly affect the retreat in terms of timing and deglaciation rate. This is because - here -

a higher winter accumulation rate helps to sustain the presence of glaciers even with warmer temperatures. Still, a delayed retreat could also be the result of an uplifted topography, which ensures temperature lapse rate near the glacier surfaces to decrease more rapidly. It would be interesting to see if the authors can reproduce similar results in the retreat either by reducing the tropospheric lapse rate (i.e. making the atmosphere cool more rapidly at increasing elevations) or by considering a synthetic higher topography (+50 m, +100 m, for example) for specific deglaciation times. Therefore, the deglaciation history might not be only defined by the applied paleo climatology, but also by climatology/topography effects that are not taken into account for the moment due to the lack in the GIA treatment. In such a case the whole discussion on climate-ice sensitivity as described in the manuscript risks becoming pointless. Put it in another words, it might be that the simulation experiment with a higher precipitation set for the whole deglaciation matches better the geological reconstructions, but for the wrong reasons.

- I understand the authors justify this deficiency as ISSM is currently lacking a GIA solver (even though they claim it's under development), and that they have already successfully applied a similar model configuration - with only relative sea level changes - in previous work in Greenland (Cuzzone et al., 2019 and Briner et al., 2020). However, to me this lack represents a big flaw in the work presented here, as I would expect that GIA has a clear primary control on the evolution of the PIS during the last deglaciation.

We think that we have adequately addressed the above comments with the clarification and addition of new text.   However, we would kindly push back a bit on the notion raised by the reviewer that GIA has a primary control on the evolution of the PIS during the last deglaciation.  GIA is a response to the ice history changes and not a forcing (Climate changes are a forcing).   Therefore, GIA cannot be a primary control on the evolution of the PIS.   With that said, GIA can modulate the response of the PIS.  GIA can influence local to regional scale elevation-mass balance feedbacks, and in turn enhance or dampen ice change.  The reviewer cites many great papers discussing current and past reconstructions of GIA across Patagonia.  Uplift rates during the deglaciation are on the order of cm/yr.  Over time, this may have some impact on mass balance through time.  However, the surface mass balance is on the order of m/yr, which is the primary control on the ice history during the deglaciation.

- Finally, the glacial outline from PATICE (Davies et al., 2020) is really uncertain in the northern part of the domain (in fact only very few radiocarbon dates were taken in the Lake district). It might be that the PIS was covering a region further north than what is presented here, as suggested by previous work (Rabassa & Clapperton 1990, Garret et al., 2020). In that case the reconstruction from Trace-21ka would not be sufficient to cover those areas and the whole discussion comparing different climate model outputs and comparing model results to the reconstructed ice retreat during the deglaciation becomes sterile. Therefore, I suggest to clearly discuss the uncertainties in the glacial reconstruction of the northern boundaries before comparing them to the model simulations.

We appreciate the reviewer highlighting the existing limitations of the terrestrial reconstruction. For this matter, we modified the text in the discussion section as it follows:

"The PATICE dataset (Davies et al., 2020) serves as the best available reconstruction of ice margin change for the PIS across the last deglaciation. This state-of-the-art compilation provides an empirical reconstruction of the configuration of the PIS as isochrones every 5ka, from 35 ka to present, based on detailed geomorphological data and available geochronological evidence. Because geochronological constraints on past PIS change are limited, the PATICE reconstruction assigns qualitative confidence to its reconstructed ice margins. Where there is agreement between geochronological and geomorphological (i.e., moraines) indicators of past ice margin history, high confidence is assigned. Where geomorphological evidence suggests the existence of past ice margins,

but lacks a geochronological constraint, medium confidence is assigned. Lastly, low confidence is assigned where there is a lack of any indicators of past ice sheet extent, where the ice limits result in interpolated interpretations from immediately adjacent moraines from valleys that have been mapped and dated. Across the CLD, the LGM ice extent is well constrained by geologic proxies particularly in the west and southwest (Figure 1). The moraines that constrain the piedmont ice lobes that formed along the western boundary are now presently lakes and have reasonable age control (Denton et al., 1999; Moreno et al., 1999; Lowell et al., 1995), giving confidence to the LGM ice margin limits. Beyond this region, age control is sparse along the western boundary for the timing of LGM ice extent, but the existence of well-defined moraines along lakes in the northern CLD are assumed to be in sync with those moraines deposited to the south (Denton et al., 1999). However, low confidence remains in the geologic reconstruction of the LGM ice boundary along the eastern margin where little to none chronological constraints are available. In general, deglaciation from the maximum LGM ice extent begins between 18 – 19 ka (Davies et al., 2020), however, poor age control and lack of geomorphic indicators make it difficult to constrain the ice extent across this region during the deglaciation. For instance, a single cosmogenic nuclide surface exposure date retrieved from the Nahuel Huapi moraine yielded an age of ~31.4 ka (Zech et al., 2017). While it is assumed that the ice limit behaved similarly both to the west and east, the limited existing data prevents a comprehensive understanding of the ice extent at the northeastern margin. This induces the highest level of uncertainty in the reconstruction and hinders our data model comparison. Therefore, we rely on the PATICE dataset interpolated isochrones (low confidence) for this northeastern region as the state-of-the-art reconstruction.”

Regarding the comparison with previous reconstructions, we argue that in fact, the PATICE reconstruction resulted in a slightly larger ice sheet extent (up to ~12%) compared to the previous state-of-the-art reconstruction (Coronato and Rabassa, 2011). Davies et al. (2020) indicate that the PATICE reconstruction is in agreement with previously published datasets, either geomorphological (Caldenius, 1932; Mercer, 1968, 1976; Coronato and Rabassa, 2011; Harrison and Glasser, 2011) or modeling studies (Hulton et al., 2002). However, when considering the Rabassa and Claperton (1990) reconstruction, it is noticeable that the northern extent of the ice sheet is in fact outside of the Patagonia region. We therefore, rely on the PATICE reconstruction for several reasons: a) it constitutes the state-of-the-art reconstruction up to date, b) relies on an extensive dataset of regional chronological constraints, c) it is consistent with topographic boundaries and geomorphological evidence, being physically plausible for ice flow dynamics. Moreover, previous developed reconstructions predate the extensive geochronological dataset currently available.

- In summary, I am afraid I cannot recommend the publication of this work in The Cryosphere until the lack of a GIA treatment in the model is either exhaustively discussed, or it is taken into account presenting a new set of simulations from the same ISSM model, when the GIA module becomes available, or from another ice-flow model which already computes the interactions between the solid Earth and the ice sheet.

We think that we have addressed the reviewer comments and clarified that we indeed account for GIA in our model simulations.

- Here I note down specific comments:
- Line 71: please describe how the SWW position and strength changed during the last deglaciation with more details.

We added this line (67):   During the LGM and last deglaciation, the position, strength, and extent of the SWW varied latitudinally, migrating southward during warmer intervals and northward during cooler intervals, ultimately altering overall ice sheet mass balance (Mercer, 1972; Denton et al., 1999; Lamy et al., 2010; Kilian and Lamy, 2012; Boex et al., 2013).

- Line 85: what do you mean by “climate?”

Changed "climate" to "precipitation"

- Line 113-115: This is already written some lines above. Please rephrase to avoid repetitions.

We adjusted line 96 to read: "To advance our understanding of last glacial and deglacial ice behavior across the CLD, we use a numerical ice sheet model to simulate the LGM ice geometry forced by an ensemble of climate boundary conditions from PMIP4 models (Kageyama et al., 2021). "

- Line 145: how is N calculated?

We have added text:

"Here $N = g(\rho_i H + \rho_w Z_b)$, where g is gravity, H is ice thickness, $\rho_I$ is the density of ice, $\rho_w$ is the density of water, and $Z_b$ is bedrock elevation following Cuffey and Paterson (2010). "

- Lines 156-161: this paragraph about the missing GIA model definitely needs further development. See my main comments above.

Please see the comments, discussion, and added text above.

- Lines 228-229: why not simulating calving at the ice-lake interface too? Could you apply the von mises stress law also there?

We have removed lines 218-229 and instead added this text and more to the Discussion (limitations) section:

"Across most of our domain, there is evidence for an advance of piedmont glaciers across glacial outwash during the LGM, which formed the physical boundary for some of the existing terminal moraines around the lakes within the CLD (Bentley, 1996; Bentley, 1997). The formation of ice-contact proglacial lakes likely occurred as a function of deglacial warming and ice retreat Bentley (1996). Where there were proglacial lakes along the westward ice front in the CLD, evidence suggests that ice was grounded during the LGM (Lago Puyehue; Heirman et al., 2011). During deglaciation, iceberg calving into the proglacial lakes may have occurred (Bentley 1996,1997; Davies et al., 2020), with evidence suggesting that local topography and calving may have controlled the spatially irregular timing of abandonment from the terminal moraines surrounding the proglacial lakes (Bentley, 1997). Recent glacier modelling (Sutherland et al., 2020) suggests that inclusion of ice-lake interactions can have large impacts on the magnitude and rate of simulated ice front retreat, as ice-lake interactions promote greater ice velocities, ice flux to the grounding line, and surface lowering. However, because the inclusion of ice-lake interactions is relatively novel for numerical ice flow modeling (Sutherland et al., 2020; Quiquet et al., 2021; Hinck et al., 2022), we choose to not model the evolution and influence of proglacial lakes on the deglaciation across this model domain. Given this limitation, our simulated magnitude and rate of ice retreat at the onset of deglaciation may be underestimated, especially when looking at local deglaciation along these proglacial lakes. Although we do not think that these processes would greatly influence our conclusions regarding the role of climate on the evolution of the PIS is the CLD region and the simulated ice retreat history, future work is required to assess the influence of proglacial lakes in this region."

- Line 248: please change "grounded" to "tidewater".

Completed

- Line 276: MIROC has drier winter conditions only in the southern part of the domain, Please correct.

Completed but please note we removed the PMIP analysis based off of Reviewer 2 comments.

- Line 399-413: I don't see the point of this sensitivity test. Yes, the experiment might be interesting to see the effect of the increased precipitation in the retreat. But what do we learn from this? Does this mean that the reconstructed precipitation is wrong? Or could this be related (also) to the missing uplift upon ice unload in your experiments? Please, discuss this further and think about other possible sensitivity tests about atmospheric lapse rate/synthetic elevation (see paragraph above).

To the GIA point, we have addressed this above and in the text.

The point of this sensitivity test follows from literature cited in our paper indicating the role of the SWW on driving changes in the hydrologic budget across the PIS and its impact on ice history. Recent ice modelling suggests the critical role of precipitation in modulating the size and extent of portions of the PIS (Muir et al., 2023; Martin et al., 2022; Leger et al., 2021). What we learn: We find that modest changes in precipitation can impact ice retreat, as we describe in the text here and the discussion section. We are very limited in understanding past climate, especially when it comes to precipitation. Few data exist reconstructing past precipitation, and that data is spatially limited. Secondly, we are limited by the fact that we only have 1 transient climate model simulation of the last deglaciation. This sensitivity test therefore raises is an important finding and signals a need for better reconstructions of past climate, including precipitation. If modest changes in precipitation can alter the surface mass balance enough to modulate ice retreat that is being driven by deglacial warming, then in order to better compare models and data in the future we also require better constraints on past climate. It is not our goal, nor did we seek, to evaluate "whether the reconstructed precipitation is wrong?"

We have added some text to the discussion section lines 532-534:

"Prior numerical ice flow modelling has indicated that precipitation played a critical role in controlling the extent of paleoglaciers of the PIS (Muir et al., 2023; Leger et al., 2021) and can modulate the retreat/advance during past intervals (Martin et al., 2022)."

Additionally, following Reviewer 2 comments we have performed 2 more sensitivity tests with different precipitation forcings. We kindly point Reviewer 1 to our response to Reviewer 2, Likewise, following Reviewer 2 comments, 3G. These simulations reinforce our conclusions regarding the ability of precipitation to modulate the pace and magnitude of deglacial ice retreat.

We have also added more text and analysis of the TraCE-21ka simulated climate as it realyes to the SWW (Please see response to Reviewer 2 section 2A). Here we have bolstered our discussion with comparison of the simulated TraCE-21ka climatology against paleoclimate reconstructions from the CLD.

- Line 450 onwards: I am missing at least a large paragraph concerning the model limitations, such as lacking GIA effect, lake-terminating calving, …, and their possible influence in the results.

Thank you. We have now added a new limitation section in the Discussion section.

*"4.3 Limitations*

Currently ISSM is undergoing model developments to include a full treatment of solid earth-ice and sea-level feedbacks (Adhikari et a., 2016). Therefore, at this time, there is no coupling between the ice sheet and solid earth. Instead, we prescribed GIA from a global GIA model of the last glacial cycle from Caron et al. (2018). While this model reasonably estimates GIA across the PIS over the last deglaciation, our simulated ice history does not feedback onto GIA. The ice history for Patagonia

incorporated into the Caron et al. (2018) ensemble is from Ivins et al. 2011. Therefore, the prescribed GIA response across our domain does not perfectly match our simulated ice history. Additionally, the global mantle from Caron et al. (2018) does not exhibit regional low viscosity that is attributable to Patagonia and therefore, current rates of deformation are likely underestimated by the model. By not simulating the 2-way coupled ice and solid-earth interactions, we could be missing some feedbacks between our simulated ice history and the solid earth that may modulate the deglaciation across this region. Despite this limitation however, our prescribed GIA from Caron et al. (2018) is reasonable when compared with reconstructed deglacial GIA in Patagonia (Troch et al., 2022), giving confidence that our simulation is capturing the regional influence of GIA on the simulated ice history.

Across most of our domain, there is evidence for an advance of piedmont glaciers across glacial outwash during the LGM, which formed the physical boundary for some of the existing terminal moraines around the lakes within the CLD (Bentley, 1996; Bentley, 1997). The formation of ice-contact proglacial lakes likely occurred as a function of deglacial warming and ice retreat Bentley (1996). Where there were proglacial lakes along the westward ice front in the CLD, evidence suggests that ice was grounded during the LGM (Lago Puyehue; Heirman et al., 2011). During deglaciation, iceberg calving into the proglacial lakes may have occurred (Bentley 1996,1997; Davies et al., 2020), with evidence suggesting that local topography and calving may have controlled the spatially irregular timing of abandonment from the terminal moraines surrounding the proglacial lakes (Bentley, 1997). Recent glacier modelling (Sutherland et al., 2020) suggests that inclusion of ice-lake interactions may have large impacts on the magnitude and rate of simulated ice front retreat, as ice-lake interactions promote greater ice velocities, ice flux to the grounding line, and surface lowering. However, across our region Heirman et al. (2011) indicate that is not well constrained how the proglacial lakes in the CLD may have influenced local deglaciation, and more geomorphic data is needed. Therefore, because the inclusion of ice-lake interactions is relatively novel for numerical ice flow modeling (Sutherland et al., 2020; Quiquet et al., 2021; Hinck et al., 2022), we choose to not model the evolution and influence of proglacial lakes on the deglaciation across this model domain. Given this limitation, our simulated magnitude and rate of ice retreat at the onset of deglaciation may be underestimated, especially when looking at local deglaciation along these proglacial lakes. Although we do not think that these processes would greatly influence our conclusions regarding the role of climate on the evolution of the PIS is the CLD region and the simulated ice retreat history, future work is required to assess the influence of proglacial lakes in this region."

- Line 478: why did you choose this "small sample" of PMIP4 climatologies? Why these models, precisely?

Please note that based off of Reviewer 2 comments, we have removed the PMIP4 analysis. We now focus on the TraCE-21ka LGM and last deglaciation experiments and have added 2 additional sensitivity tests as described above and in our response to Reviewer 2 (section 3G).

- Lines 528: again, could not this be related to the missing regional uplift too?
As described above GIA is prescribed in our model simulations.

- Lines 557-565: I suggest to work on these conclusions as they are only partly corroborated by your sensitivity test.

Given that we do indeed include GIA in our simulations and after clarifying in text, we think our original conclusions stand.

Figures:

- Figure 1, figure 2: it would be helpful to add some reference locations to the map (e.g. gulf of Ancud, Seno de Reloncaví, …) and lat/lon.

We have added this to Figure 2.

[Figure]

Figure 1. Bedrock topography for our study area (meters). Our model domain (shown as the black line), encompasses the reconstructed LGM ice limit (shown in red) from PATICE (Davies et al., 2020). Present day lakes are shown in blue, with abbreviated names as: SR (Seno de Reloncaví), GA (Golfo de Ancud), LL (**Lago Llanquihue), LR1 (Lago Rupanco)**, LP1 (Lago Puyehue), LR2 (Lago Ranco), LR3 (Lago Riñihue), LP2 (Lago Panguipulli), LC (Lago Calafquén), LV (Lago Villarica), LNH (Lago Nahuel Huapi).

- Figure 3: I would like to see a further discussion about the reasons that explain the main differences between the PMIP4 models (model parametrisations, …) or at least citing some papers that point to that.

We do not think the burden is on us to discuss the differences in the model parameterizations as work is still being done to evaluate these models, and the role of each models' individual parameterizations on the results is often not evaluated in detail. Main goals of the PMIP experiments are captured by Kageyama et al., 2017 (See Section 2). Often large model differences can be down to differences in climate sensitivity, which may be related to things such as cloud feedbacks. We can add the citation

from Brierley et al., 2020, which has some text about possible reasons for differences in simulated climate.

Additionally, we have removed the discussion of the PMIP4 simulations per Reviewer 2 comments, and instead focused on more TraCE-21ka sensitivity experiments and last deglacial experiments.

Citation: Masa Kageyama, Samuel Albani, Pascale Braconnot, Sandy Harrison, Peter Hopcroft, et al.. The PMIP4 contribution to CMIP6 – Part 4: Scientific objectives and experimental design of the PMIP4- CMIP6 Last Glacial Maximum experiments and PMIP4 sensitivity experiments. Geoscientific Model Development Discussions, Copernicus Publ, 2017, 10 (11), pp.4035-4055. ff10.5194/gmd-10-4035-2017ff. ffhal-02328464f

Brierley, C. M., Zhao, A., Harrison, S. P., Braconnot, P., Williams, C. J. R., Thornalley, D. J. R., Shi, X., Peterschmitt, J.-Y., Ohgaito, R., Kaufman, D. S., Kageyama, M., Hargreaves, J. C., Erb, M. P., Emile-Geay, J., D'Agostino, R., Chandan, D., Carré, M., Bartlein, P. J., Zheng, W., Zhang, Z., Zhang, Q., Yang, H., Volodin, E. M., Tomas, R. A., Routson, C., Peltier, W. R., Otto-Bliesner, B., Morozova, P. A., McKay, N. P., Lohmann, G., Legrande, A. N., Guo, C., Cao, J., Brady, E., Annan, J. D., and Abe-Ouchi, A.: Large-scale features and evaluation of the PMIP4-CMIP6 *midHolocene* simulations, Clim. Past, 16, 1847–1872, https://doi.org/10.5194/cp-16-1847-2020, 2020.

- Figure 6: why not plotting the same figure with respect to the glaciated area and comparing it to PATICE reconstruction? It could be also interesting to plot the same figure, but separately for the northern and southern parts of the domain (e.g. north and south of 40°S) since climatologies present a strong latitudinal pattern.

We have removed the discussion of the PMIP4 simulations per Reviewer 2 comments, and instead focused on more TraCE-21ka sensitivity experiments and last deglacial experiments.

- Figure 7: please choose a different color scale as ice lost from the LGM to 17 ka ago is very difficult to see.

We have changed the colors and hope this is better visually. We note however, that following Reviewer 2 we have changed up the format and will only show Figure A below. In our sensitivity test section (3.2.2) we have added 3 more panels showing the difference in simulated deglaciation age between our standard run (Figure A below) and 3 new simulations that test the sensitivity of ice retreat to precipitation scenarios. We would direct Reviewer 1 to our response to Reviewer 2 (section 3G).

[Figure]

- 10: please use different colours for the simulated outlines (orange, red?) otherwise they can be used with the topography.

Below is an example of new colormaps for the figure. We hope you find this easier to read.

[Figure]

References:

Bourgois, J., et al., 2016. Geomorphic records along the general Carrera (Chile)-Buenos Aires (Argentina) glacial lake(46°-48°S), climate inferences, and glacial rebound for the past 7-9 ka. J. Geol.124, 27e53. https://doi.org/ 10.1086/684252.

Briner et al., 2020, Rate of mass loss from the Greenland Ice Sheet will exceed Holocene values this century. 2020. Nature, 6, 70–74, https://doi.org/10.1038/s41586-020-2742-6.

Cuzzone et al., 2019, The impact of model resolution on the simulated Holocene retreat of the southwestern Greenland ice sheet using the Ice Sheet System Model (ISSM). 2019. The Cryosphere, 13, 879–893, https://doi.org/10.5194/ tc-13-879-2019.

Davies et al., 2020. The evolution of the Patagonian Ice Sheet from 35 ka to the present day (PATICE). Earth Sci. Rev. 204, 103152. https://doi.org/10.1016/j.earscirev.2020.103152.

Dietrich, R., et al. "Rapid crustal uplift in Patagonia due to enhanced ice loss." Earth and Planetary Science Letters 289.1-2 (2010): 22-29.

Jara-Mu~noz, J., Melnick, D., 2015. Unravelling sea-level variations and tectonic uplift in wave-built marine terraces, Santa María Island, Chile. Quat. Res. 83, 216e228

Lange, H., et al. "Observed crustal uplift near the Southern Patagonian Icefield constrains improved viscoelastic Earth models." Geophysical Research Letters 41.3 (2014): 805-812.

Rabassa, Jorge, and Chalmers M. Clapperton. "Quaternary glaciations of the southern Andes." Quaternary Science Reviews 9.2-3 (1990): 153-174.

Richter, A., et al., 2016. Crustal deformation across the southern patagonian icefield observed by GNSS. Earth Planet Sci. Lett. 452, 206e215. https://doi.org/10.1016/j.epsl.2016.07.042.

Ríos, F., et al., 2020. Environmental and coastline changes controlling Holocene carbon accumulation rates in fjords of the western Strait of Magellan region. Continent. Shelf Res. 199, 104101. https://doi.org/10.1016/j.csr.2020.104101

Stefer, S., et a., 2010. Forearc uplift rates deduced from sediment cores of two coastal lakes in south-central Chile.
Tectonophysics 495, 129e143.

Troch, Matthias, et al. "Glacial isostatic adjustment near the center of the former Patagonian Ice Sheet (48° S) during the last 16.5 kyr." Quaternary Science Reviews 277 (2022): 107346.

---

## Author Comment (AC2)

**Please note that the Reviewer Comments are posted in Black, Author Responses posted in Red, and our proposed Revisions to the text posted in Blue.**

- **The manuscript is presenting reconstructions of the northern sector of the former Patagonian ice sheet during the last glacial maximum (LGM) and subsequent deglaciation based on an ice sheet model ensemble driven by the climate model forcing from the PMIP4 and TRACE-21ka experiments. One shortcoming of this manuscript is an obvious disconnect between the LGM and deglaciation experiments – it almost feels like they belong to two separate studies. If this manuscript is to be published, the synergy between these two parts of the study must be improved considerably. There are also several issues with the methodology that adapts poorly justified assumptions and simplifications requiring much more detailed considerations and sensitivity tests. Finally, the interpretation of the large-scale processes driving regional climate changes is weak, and the link between climate models and paleoclimate proxy data is non-existent. Below I provide detailed instructions for major revisions of the study's design and contents that are required to merit a publication in TC.**

We would like to thank the reviewer for their consideration and generous suggestions as to improve the analysis and conclusions expressed in this paper. When considering their comments, we have decided to make some changes to our analysis. Based on new work that is in discussion at Climates of the Past (https://cp.copernicus.org/preprints/cp-2023-47/) which takes a deeper look into the PMIP4 climatologies and simulated LGM PIS behavior (in addition to Yan et al., 2020), we have, based upon the reviewer suggestions, to remove the PMIP analysis from this paper. Instead, and upon further analysis suggested by the reviewer, we have decided to focus on the LGM and last deglaciation simulations using TraCE-21ka climate forcings. To assess the role of the SWW and the influence of precipitation, we have performed two additional transient ice sheet simulations across the last deglaciation – see response to Reviewer Comment 3G below. We find that these new simulations provide stronger support to our initial conclusions regarding the modulating role precipitation may have played on influencing the pace and magnitude of deglaciation across our model domain.

We also thank the reviewer for the gentle nudge to look deeper into the deglacial climate changes simulated within the TraCE-21ka climate model. Below we highlight additional analysis of the TraCE-21ka model outputs across South America and our model domain. This analysis shows that TrACE-21ka simulates changes in the SWW position and strength, and ultimately the hydrologic cycle during the LGM and last deglaciation across the CLD. Additionally, we find that the TraCE-21ka simulated deglacial changes in precipitation and SWW qualitatively agree well with paleoclimate proxies of precipitation across the CLD, from which prior worked has used to infer deglacial changes in the SWW. We link this discussion to our ice sheet model results and provide more detailed commentary comparing simulated changes in TrACE-21ka precipitation and temperature to proxy records across the Chilean Lakes District that constrain deglacial climate change. From this analysis, we hope it is more clear that TraCE-21ka compares well against regional proxies of past climate change across the CLD, giving confidence that our ice sheet modelling is capturing changes consistent with the prevailing understanding of deglacial climate change in this region.

We respond to the other points raised by the reviewer by offering results of additional modelling experiments and clarifying our text to meet the reviewers' comments.

**Major points:**

- **Disconnect between two parts of the study: Both the results and discussion sections (especially the latter) have very weak links between the LGM and deglaciation components of the study. It is unclear why there is a need for the PMIP4-driven LGM experiments if they don't contribute much to the design of the transient simulations. More effort needs to be invested into strengthening the synergy through cross-model experiments and large-scale mechanism interpretation.**

- **2. Weak interpretation of the large-scale mechanisms:**
- **2a) The interpretation of the origins of temperature and precipitation anomalies and their impacts on the ice sheet formation and growth is weak. There are a lot of mentions of the southern westerly wind (SWW) system as a potential driver and yet no attempt to look into the climate model outputs and establish to which extent this process is a factor. The statements in the manuscript that paleoclimate proxies provide a confusing picture are not helpful and hint at an energy-saving mode of the study (minimum effort). Having global climate model outputs at hand, the authors need to step up and dive into the simulated climate dynamics and large-scale drivers of the inferred anomalies.**

Thank you for the push to look closer at local paleoclimate proxies of climate change across the CLD. We have bolstered the review of current literature surrounding paleoclimate change across the CLD, and the nature of how the SWW is interpreted to have changed based upon these proxy records. We have made changes to our introductory text and the discussion. In the discussion section we include more information that provides a qualitative comparison between TraCE-21ka simulated temperature and precipitation change across the CLD and information derived from proxies (e.g. vegetation, pollen).

We kindly push back on what the reviewer stated regarding, "The statements in the manuscript that paleoclimate proxies provide a confusing picture are not helpful and hint at an energy-saving mode of the study (minimum effort)." We would like to clarify that we never mentioned the paleo proxies being "confusing," nor was that our intention. We apologize for any miscommunication on our part.

Instead, we merely cited evidence from the literature to support the notion that it is very difficult to constrain SWW changes directly from paleo-proxy data. Perhaps this is best summarized in Kohfeld et al., 2013 who looked at a large dataset of paleoclimate proxies across the Southern Hemisphere (many over S. America) and concluded: "A chain of assumptions are needed to interpret paleodata as changes in the westerly wind position and intensity. Importantly, the modern relationships between Southern Hemisphere westerly winds and moisture must hold for past time

periods, and over an increased latitudinal range at the LGM, compared with the present day…..These inherent assumptions, and the possibility for multiple interpretations of the observations, create uncertainty in conclusions regarding past changes in winds interpreted solely from data."

This is also supported by additional climate modelling work from Sime et al., 2013, whom Kohfeld state: "Combining these data compilations with model simulations is one approach for better understanding the role of winds in controlling glacial interglacial conditions. Sime et al. (2013) use this moisture reconstruction to assess impacts of LGM wind fields on moisture patterns from several AGCM and AOGCM simulations. Their results suggest that model simulations do a reasonable job of reproducing LGM moisture patterns without large shifts in glacial winds and provide one example of integrating model simulations with data compilations to understand ocean-atmosphere changes during the LGM."

Therefore, we have reworked our introduction and discussion to clarify any misunderstanding we may have created with our assessment.

Introduction (Lines 71: 93), we have updated and added text accordingly:

"Terrestrial paleoclimate proxies indicate that the CLD was wetter during the LGM and early deglaciation, supporting the idea that the SWW migrated northward of 41°S across the CLD (Moreno et al., 1999; Moreno et al., 2015; Moreno and Videla, 2016). Additionally, these proxies indicate a switch from hyperhumid to humid conditions around 17,300 cal yr BP, which was inferred by Moreno et al. (2015) to indicate the poleward migration of the SWW south of the CLD. However, we note that inferring changes in the SWW across the last deglaciation from paleoclimate proxies can be problematic as outlined by Kohfeld et al. (2013) who compile an extensive dataset of proxy records that record changes in moisture, precipitation-evaporation balance, ice accumulation, runoff and precipitation, dust deposition, and marine indicators of sea surface temperature, ocean fronts, and biologic productivity across the Southern hemisphere. Kohfeld et al. (2013) conclude that environmental changes inferred from existing paleoclimate data could be potentially explained by a range of plausible scenarios for the state and change of the SWW during the LGM and last deglaciation, such as a strengthening, poleward or equatorward migration, or no change in the SWW. Climate model results from Sime et al. (2013) indicate that the reconstructed changes in moisture from Kohfeld et al. (2013) can be simulated well without invoking large shifts or changes in strength to the SWW. This discrepancy also exists amongst climate models which diverge on whether the LGM SWW was shifted equatorward or poleward, and was stronger or weaker (Togweiler et al., 2006; Menviel et al., 2008; Rojas et al., 2009; Rojas et al., 2013; Sime et al., 2013; Jiang et al., 2020). Therefore, from paleoclimate proxies and climate models, we still do not have a firm understanding of how the SWW may have changed during the last deglaciation, however, climate proxies and models can still be effectively used to evaluate regional or local climate changes despite uncertainty in the dynamical cause."

We note that it is difficult to tie precipitation changes across our domain and South America to one specific dynamical forcing such as changes in the SWW. To do so would require a separate analysis and dedication to another paper. However, we have added some additional analysis, which we will add to a supplement, regarding how TraCE-21ka simulates changes in large atmospheric

scale circulation and associated changes in the SWW.  We analyzed outputs of 925 hPa zonal winds and computed moisture flux convergence to help fill gaps in our current manuscript.  Below is text that will be added to our Discussion section 4.1.

** Please note:  We have attached figures at the end of this document which we cite here in our response.  These figures will be added to a supplement.

[revised manuscript text omitted]

W.R. Peltier, 2004. Global Glacial Isostasy and the Surface of the Ice-Age Earth: The ICE-5G (VM2) Model and GRACE, Ann. Rev. Earth and Planet. Sci., 32, 111-149.

- **2b) It remains unexplained what drives the increased winter precipitation in TRACE-21ka between 22 and 18 ka relative to the preindustrial (PI). Are the inferred driving mechanisms supported by at least one of the PMIP4 models? It is also unclear why the inferred deviation between winter precipitation anomalies in the north and south disappears after 16 ka. Finally, what mechanisms drive dips is the winter precipitation relative to the PI after 16 ka?**

We would like to refer the reviewer to our discussion in 2a. We have done more analysis of the TraCE-21ka outputs and with existing literature. The discussion provides more detail on the possible controls on the simulated changes in precipitation during these time intervals. Likewise, we have added additional context as to how the changes simulated in TraCE-21ka relate to the paleoclimate records across the CLD.

- **2c) While the idea of using TRACE-21ka for the applications in Patagonia is new, this climate forcing suffers from a very low spatial resolution (T31) that is suboptimal for a region with such steep topography and its regional performance has not been validated against any paleoclimate proxy data in South America or neighboring areas. The lack of validation and interpretation of the inferred climate time series from TRACE-21ka against existing sediment and ice core records and other proxies, both regional and semi-hemispheric, is a flaw of the study. It is incorrect that such reconstructions are missing in the region, and at least some effort could be made to compare with the signals reconstructed from the West Antarctic ice core data.**

Large, multi-proxy reconstructions from papers by He and Clark 2022, Liu et al. 2009, He et al. 2011, Shakun et al. 2015, Shakun et al. 2012, Marcott et al. 2013, etc. have all demonstrated good agreement between TRACE 21k and paleo proxy data. These studies include a wide variety of paleo-proxy data that include West Antarctic and proxy records from South America. Our own analyses (unpublished – see figure below of temperature reconstruction) has also show generally good agreement between TRACE 21k and existing records from the last glacial to present in the Southern Hemisphere. Additionally, we have now undertaken analyses in this paper (see Discussion above 2A) which further demonstrates the agreement between the modeling results and local paleo-precipitation and temperature proxies.

[Figure]

- **3. Methodology flaws and missing information: There are quite some simplifications and limitations in the methodology and experimental design that must be considered in a greater detail.**
- **3a) The impacts of the missing treatment of ice temperature and viscosity on the ice sheet dynamics must be demonstrated as negligible. Currently it is dismissed as a factor through weak arguments. The computational efficiency may be a good reason for model simplifications but not at the cost of the non-physical model outcomes. I see the need for sensitivity experiments that would quantify the impacts and related uncertainties in the study's conclusions. Also, a reference supporting the statement in lines 135-136 is painfully missing, including the considerations that in areas with such thin lithosphere and high geothermal flux, temperate basal conditions do not always translate into vertically temperate glacier regimes.**

In order to test the validity of our assumption of a largely temperate based ice sheet across this domain, we calculate ice temperature at the LGM, assuming the ice sheet is in a steady-state thermal equilibrium following Serrousi et al. (2013).   This methodology has been used for numerous applications in Greenland and Antarctica to calculate the thermal conditions of the ice sheets (Seroussi et al., 2013; MacGregor et al., 2016; Goelzer et al., 2020; Seroussi et al., 2020). We use our modeled LGM ice sheet state for the SynTraCE-21ka simulation (ice sheet geometry) to calculate the thermal conditions of the ice sheet.  This formulation, outlined in Seroussi et al. (2013) uses an enthalpy formulation from Aschwanden et al. (2012) that includes both temperate and cold ice.  LGM air temperature is imposed at the surface and geothermal heat flux is applied at the base (100 mW m−2 mean from Hamza and Vieira, 2018).  For this step, we extrude our 2D model to 3D, with the model for the thermal state calculation having 20 layers.

Here we show the simulated Steady State Basal Temperatures (A) and the simulated Depth Averaged Steady State temperatures (B). See below for figures.

The ice sheet is mainly warm based (figure A; below) with temperatures near 0 degrees Celsius, with exception for some of the high peaks where ice cover is thin enough for the colder surface temperatures to diffuse and advect downward. The depth averaged temperature (figure B; below) shows that the majority of the ice sheet is between -1 to 0 degrees Celsius, with the mean of the depth averaged temperature being -0.41 degrees Celsius. There are some exceptions, where colder ice seems to be advected downstream from the colder based high peaks (in figure A), however, we must note that these simulated temperatures are likely an underestimate as they do not account fully for frictional heating that would occur if the 3D thermal model was run transiently (mass transport and stress balance) to steady state.

Therefore, based on this additional analysis, we think that our assumption in setting the ice temperature in our 2D model to -0.2 degrees Celsius is justified. We will add this analysis and text to the supplement and reference it in our text discussion on the thermal state of the ice sheet (section 2.1 Ice Sheet Model).

We have added a reference in lines 140-141 and adjusted the text to reflect this:

"Although geomorphological evidence suggests that while southernmost glaciers across the PIS may have been temperate with warm based conditions during the LGM, there may have been periods where ice lobes were polythermal (Darvill et al., 2016). However, recent ice flow modelling (Leger et al., 2021) suggests that varying ice viscosity mainly impacts the accumulation zone thickness in simulations of paleoglaciers in Northeastern Patagonia, with minimal impacts on overall glacier length and extent. Accordingly, based on sensitivity tests (see supplement), our model is 2-dimensional and we do not solve for ice temperature and viscosity allowing for increased computational efficiency."

We additionally note that geomorphological and topographic evidence is suggestive of widespread temperate ice conditions across the north Patagonian batholith. This is depicted in abraded surfaces at different elevations ranging from 900-2000 m.a.s.l, including the highest peaks. Glacially polished bedrock surfaces were recognized through ground-validating fieldwork (Romero et al., 2023 INQUA Abstract), suggesting that the ice sheet could have been at the pressure-melting point at its base, ruling out signs of cold based conditions. Moreover, results from thermochronology studies revealed that higher latitudes in Patagonia experienced reduced late Cenozoic erosion in comparison to northern sites (Thomson et al., 2010). This study noted that northern sites of the Patagonian Andes did not exhibit signs of glacial protection from erosion, indicating that southern sites must have experienced spatially restricted glacier basal flow. This kind of glacial protection of the landscape has been determined to occur in polar regions mostly. Therefore, we suggest that geomorphic evidence and thermochronological studies highlight that northern Patagonia would have exhibited temperate ice conditions.

[Figure]

*Figure A. Simulated Steady State basal temperature in degrees Celcius.*

[Figure]

*Figure B. Simulated depth averaged ice temperature in degrees Celcius.*

**References:**

Aschwanden, A., Bueler, E., Khroulev, C., and Blatter, H.: An enthalpy formulation for glaciers and ice sheets, J. Glaciol., 58, 441–457, https://doi.org/10.3189/2012JoG11J088, 2012.

Darvill, C.M., Stokes, C.R., Bentley, M.J., Evans, D.J.A., Lovell, H.  Dynamics of former ice lobes of the southernmost Patagonian Ice Sheet based on a glacial landsystems approach.  Journal of Quaternary Science. 32,6,857-876. https://doi.org/10.1002/jqs.2890

Goelzer, H., Nowicki, S., Payne, A., Larour, E., Seroussi, H., Lipscomb, W. H., Gregory, J., Abe-Ouchi, A., Shepherd, A., Simon, E., Agosta, C., Alexander, P., Aschwanden, A., Barthel, A., Calov, R., Chambers, C., Choi, Y., Cuzzone, J., Dumas, C., Edwards, T., Felikson, D., Fettweis, X., Golledge, N. R., Greve, R., Humbert, A., Huybrechts, P., Le clec'h, S., Lee, V., Leguy, G., Little, C., Lowry, D. P., Morlighem, M., Nias, I., Quiquet, A., Rückamp, M., Schlegel, N.-J., Slater, D. A., Smith, R. S., Straneo, F., Tarasov, L., van de Wal, R., and van den Broeke, M.: The future sea-level contribution of the Greenland ice sheet: a multi-model ensemble study of ISMIP6, The Cryosphere, 14, 3071–3096, https://doi.org/10.5194/tc-14-3071-2020, 2020.

Seroussi, H., Nowicki, S., Payne, A. J., Goelzer, H., Lipscomb, W. H., Abe-Ouchi, A., Agosta, C., Albrecht, T., Asay-Davis, X., Barthel, A., Calov, R., Cullather, R., Dumas, C., Galton-Fenzi, B. K., Gladstone, R., Golledge, N. R., Gregory, J. M., Greve, R., Hattermann, T., Hoffman, M. J., Humbert, A., Huybrechts, P., Jourdain, N. C., Kleiner, T., Larour, E., Leguy, G. R., Lowry, D. P., Little, C. M., Morlighem, M., Pattyn, F., Pelle, T., Price, S. F., Quiquet, A., Reese, R., Schlegel, N.-J., Shepherd, A., Simon, E., Smith, R. S., Straneo, F., Sun, S., Trusel, L. D., Van Breedam, J., van de Wal, R. S. W., Winkelmann, R., Zhao, C., Zhang, T., and Zwinger, T.: ISMIP6 Antarctica: a multi-model ensemble of the

Antarctic ice sheet evolution over the 21st century, The Cryosphere, 14, 3033–3070, https://doi.org/10.5194/tc-14-3033-2020, 2020.

MacGregor, J. A., Fahnestock, M. A., Catania, G. A., Aschwanden, A., Clow, G. D., Colgan, W. T., Gogineni, S. P., Morlighem, M., Nowicki, S. M., Paden, J. D., and Price, S. F.: A synthesis of the basal thermal state of the Greenland Ice Sheet, J. Geophys. Res.-Earth, 121, 1328–1350, 2016.

Romero, M., Cuzzone, J.K., Jones, A.G., Bushmaker, S., Marcott, S.A. Post-Glacial dynamics of the Patagonian Ice Sheet across the Southern Volcanic Zone. In: XXI INQUA Congress, July 14ᵗʰ – 20ᵗʰ 2023, Sapienza University of Rome, Italy

Seroussi, H., Morlighem, M., Rignot, E., Khazendar, A., Larour, E., & Mouginot, J. (2013). Dependence of century-scale projections of the Greenland ice sheet on its thermal regime. Journal of Glaciology, 59(218), 1024–1034. https://doi.org/10.3189/2013JoG13J054

Hamza, V. M. and Vieira, F.: Global heat flow: new estimates using digital maps and GIS techniques, International Journal of Terrestrial Heat 490 Flow and Applied Geothermics, 1, 6–13, https://doi.org/10.31214/ijthfa.v1i1.6, 2018

Thomson, Stuart N., et al. "Glaciation as a destructive and constructive control on mountain building." *Nature* 467.7313 (2010): 313-317.

- **3b) It must be at least discussed how the absence of the Glacial Isostatic Adjustment (GIA) modeling is impacting the model reconstructions and conclusions of this study. While I do not fully agree with the referee 1 that the GIA is a controlling factor, given the high uncertainties in the model parameters, parameters of the downscaling procedure and external forcings, I agree with their arguments that this limitation of the model must be addressed in a much more thorough manner. Here I refer the authors to the detailed suggestions of the referee 1.**

We would kindly point Reviewer 2 to our response to Reviewer 1 where we address these concerns and revisions in full.

- **3c) Given the steep topographic gradients that dominate regional climate, it is not discussed enough how the choice of such a small model domain compared to the grid sizes of the climate models is impacting the outcomes of the modeling experiments, hugely inflating the role of the chosen parameters for downscaling.**

Our model domain encompasses 4 TrACE-21ka gridcells. While the climate model outputs cannot adequately capture the steep topographic gradients as expressed by the reviewer, we apply anomalies of climatologies from the climate models and not the raw data. The anomalies are applied onto a high-resolution reanalysis product (CR2MET), and the temperature and precipitation are adjusted following lapse rate adjustments (temperature) and elevation desertification (for precipitation). Please see comment 3E below. We apply a standard modeling approach (Pollard et al., 2012; Seguinot et al., 2016; Golledge et al., 2017; Tigchlaar et al., 2019; Clark et al., 2020; Briner et al., 2020; Cuzzone et al., 2022; Yan et al., 2022): using climate model

output allows us to capture more spatial variability that accounts for changes in climate due to large scale atmospheric circulation change versus, for instance, ice core scaling techniques which only rely on scaling a present day climatology based on variations from a far field site. While perfectly acceptable, ice core scaling techniques often take remote ice cores (often from Greenland or Antarctica) changes in temperature and apply these anomalies to a large geographical area; for example, see Seguinot et al., 2016 (citation below in 3E). While it may be more beneficial to have higher resolution climate model output that accounts for higher resolution topography, our approach is sound and still allows for an improved understanding of the possible deglaciation of the PIS across this region especially given the large uncertainty in the geologic reconstruction during the later deglaciation. Additionally, thanks to the Reviews suggestion to compare TraCE-21ka to regional proxies in the CLD, we are more confident that changes simulated in TraCE-21ka match well with the proxy records.

Likewise, we do not feel we have inflated the role of chosen parameters. For the transient (last deglaciation) simulations, we need to initialize our model. This requires some parameter choice such that the simulated LGM ice sheet fits some "observed" state. In this case (as described in comment 3D below), our simulated LGM ice sheet using the TraCE-21ka climate has a good fit to the PATICE LGM ice area (within 5%; figure 11), and therefore we can say that the model achieves a good fit to the "observed" LGM extent from PATICE. We then keep those parameters constant and apply the deglacial climate changes to simulate the last deglaciation across our model domain. This approach is similar to how the community simulates future or past ice sheet change for glaciers or Greenland/Antarctica. Parameter choices need to be made that satisfy an initial simulated ice state against some observable, giving confidence that the simulated transient changes are robust.

- **3d) Referring to Yan et al.'s paper for the choice of internal model parameters is a poor practice since this paper utilizes model parameters that are fine-tuned to support their desired outcomes rather than parameters supported by observational evidence from glaciated regions on our planet today. This brings us to a question – how have the choices of lapse rate and PDD model parameters impacted the modeled extents and volumes of the ice sheet?**

We kindly point the reviewer to a citation we also listed (Fernandez et al., 2016) in text. They list published model parameter choices for the positive degree day approach as it applies to modeling contemporary and historical glacier change across South America. Parameter values vary between studies, however, the values we have chosen for this study are within the range of values used across this region, and are therefore supported by observational and modeled evidence from glaciated regions in South America.

Because our transient ice modelling across the last deglaciation uses the TrACE-21ka climate outputs, our degree day and lapse rate parameters were chosen so that the simulated LGM ice area was in close agreement with the reconstructed PATICE area at the LGM (within 5% of reconstructed ice area for PATICE; Figure 11). This provides confidence in our modeled LGM state using the TrACE-21ka climate outputs. And for the transient deglacial simulations, these parameters remain constant and only the climate forcing varies through the last deglaciation simulation. Again, we follow standard practice in ice sheet modelling: that is, once a model is

initialized, and parameters are chosen that provide a good match between a modeled and observed LGM state, those parameters are held constant during transient simulations.

We can add text in Section 2.3 Surface Mass Balance (line 223), to reflect these choices in model parameters: "Using these parameter values, we arrive at a simulated LGM ice sheet area matches well (within 5%) to the reconstructed ice area from PATICE (see Figure 11) for the simulation using the TraCE-21ka climate forcing."

- **3e) The resulting ice sheet geometries presented in Figure 5 in response to different PMIP4 climate forcings and in combination with the listed Positive Degree Day (PDD) parameters are surprising, to say the least. Both AWI and MPI climate model outputs must be modified significantly to enable the growth of an ice sheet in this area. I am missing the information about the model parameters adapted in this study – for example, what is the daily temperature standard deviation in the PDD model? This is a principal parameter that shapes glacier and snow melt. A more detailed description of the downscaling procedure is also needed. For example, how was precipitation downscaled/corrected? How was the ice sheet boundary forcing in the climate models accounted for when downscaling? A table in the appendix with all major model parameters and a detailed method description would be much appreciated.**

We have added this to section 2.3: "The hourly temperatures are assumed to have a normal distribution, of standard deviation 3.5 degrees Celsius around the monthly mean." Our other mass balance parameters are listed in that section but if a table is easier to digest, than we will be more than happy to include one.

We discuss the application of the climate forcings in Section 2.4. We use a common approach in numerical ice flow modeling to simulate ice behavior across paleoclimate timescales. We point the reviewer to Section 2.4.

We have added additional citations in Section 2.4 to support this: "In order to scale monthly temperature and precipitation across the LGM and last deglaciation we applied a commonly used modeling approach (Pollard et al., 2012; Seguinot et al., 2016; Golledge et al., 2017; Tigchlaar et al., 2019; Clark et al., 2020; Briner et al., 2020; Cuzzone et al., 2022; Yan et al., 2022)" As described in the text, the lapse rate is used to down scale the temperatures onto the resulting topography (ice surface).

To scale precipitation an elevation-dependent desertification is included (Budd and Smith, 1981) which reduces precipitation by a factor of 2 for every kilometer change in ice sheet surface elevation. We have added this text to section 2.4: "An elevation-dependent desertification is included (Budd and Smith, 1981) which reduction in precipitation by a factor of 2 for every kilometer change in ice sheet surface elevation (Budd and Smith, 1981) ."

We acknowledge that not including calving on proglacial lakes represents limitation in our deglacial simulations and we have added text to a new Limitations section (4.3) in the Discussion.

We have added some text in the Limitations section (Section 4.3).

"Across most of our domain, there is evidence for an advance of piedmont glaciers across glacial outwash during the LGM, which formed the physical boundary for some of the existing terminal moraines around the lakes within the CLD (Bentley, 1996; Bentley, 1997). The formation of ice-contact proglacial lakes likely occurred as a function of deglacial warming and ice retreat Bentley (1996). Where there were proglacial lakes along the westward ice front in the CLD, evidence suggests that ice was grounded during the LGM (Lago Puyehue; Heirman et al., 2011). During deglaciation, iceberg calving into the proglacial lakes may have occurred (Bentley 1996,1997; Davies et al., 2020), with evidence suggesting that local topography and calving may have controlled the spatially irregular timing of abandonment from the terminal moraines surrounding the proglacial lakes (Bentley, 1997). Recent glacier modelling (Sutherland et al., 2020) suggests that inclusion of ice-lake interactions may have large impacts on the magnitude and rate of simulated ice front retreat, as ice-lake interactions promote greater ice velocities, ice flux to the grounding line, and surface lowering. However, across our region Heirman et al. (2011) indicate that is not well constrained how the proglacial lakes in the CLD may have influenced local deglaciation, and more geomorphic data is needed. Therefore, because the inclusion of ice-lake interactions is relatively novel for numerical ice flow modeling (Sutherland et al., 2020; Quiquet et al., 2021; Hinck et al., 2022), we choose to not model the evolution and influence of proglacial lakes on the deglaciation across this model domain. Given this limitation, our simulated magnitude and rate of ice retreat at the onset of deglaciation may be underestimated, especially when looking at local deglaciation along these proglacial lakes. Although we do not think that these processes would greatly influence our conclusions regarding the role of climate on the evolution of the PIS is the CLD region and the simulated ice retreat history, future work is required to assess the influence of proglacial lakes in this region. "

- **3g) The design of the sensitivity analysis is incomplete, further emphasizing the disconnect between the LGM and transient experiments. This is an area where stronger connections between the two sets of experiments can be built by for example, also testing precipitation anomalies from at least some of the utilized PMIP4 models. It would be also beneficial to test how the deglaciation would look like if the PI precipitation and present-day observed precipitation rates were used to drive the deglaciation run. It would allow this study to decouple the impacts of higher-than-PI winter precipitation prior to 18 ka and lower-than-PI winter precipitation after 16 ka the pace of the deglaciation.**

We agree with the reviewer that additional sensitivity tests are needed to better support the role of precipitation in modulating deglacial ice retreat across the CLD. Since we decided to remove the PMIP4 simulations from this manuscript, we added 2 additional simulations using the TraCE-21ka model output. In the current manuscript we have 2 transient simulations. One where the TraCE-21ka climate boundary conditions vary through time (listed as #1 below and referred to as the **main simulation**) and a second simulation where we fix the precipitation during the transient simulation to be equivalent to the monthly mean from 22ka to 20ka (#4 below). Taking the

reviewers suggestion, we conduct 2 more transient simulations (#2 and #3 below).  In all of these sensitivity experiments (#2-4 below and referred to as sensitivity experiments 2-4), temperature varies across the last deglaciation but precipitation remains fixed at the given magnitude for a particular chosen time interval.

List of simulations:

1) Climate boundary conditions (temperature and precipitation) vary through time (denoted as the **main simulation**).
2) Monthly precipitation is held constant at the preindustrial mean.  This is what the reviewer suggests above.
3) Monthly precipitation is held constant at the mean 12.5 ka-12 ka values.  From figure 9 in the current manuscript, we see that this is a period of reduced precipitation relative to the preindustrial (~7% reduction).
4) Transient simulation where the monthly precipitation is held constant to the 22-20 ka mean from TraCE-21ka, which is roughly 10% higher than preindustrial values across the Northern portion of the model domain (North of 40°S).

As the reviewer suggests, sensitivity experiments 2 and 4 allows us to better assess the impacts of higher than PI winter precipitation prior to 18ka, while experiment 3 allows us to assess how a reduced precipitation during the deglaciation influences the overall pacing of ice retreat.

The results are shown below as the difference in the simulated deglaciation between the sensitivity experiments (#2,3,4) and the main simulation (#1).  The blue colors indicate that the simulated deglaciation for the sensitivity experiments are slower than the main simulation, and the red colors indicate that the simulated deglaciation for the sensitivity experiments are faster than the main simulation.

In Figure SA below, the difference in the deglaciation age between sensitivity experiment 2 and the main simulation is shown, where precipitation is held constant at the preindustrial value. Please note we call this Figure S, but will give it an appropriate Figure # in the revised manuscript.  Across our model domain, wintertime precipitation during the preindustrial is reduced compared to the early deglaciation (22 ka to 18ka) and is similar to slightly higher particularly south of 40°S after 18 ka (Figure 9 from current manuscript).  When holding precipitation constant at the preindustrial value through the last deglaciation, we notice that ice retreats faster across most portions of the model domain, particularly along the ice margins and in the northern sector.  In the southern portion of our model domain, where the relative changes in deglacial precipitation relative to the preindustrial are smaller, the difference in simulated deglaciation age are also smaller.  In general, the pace of deglaciation increases by up to 1 kyr compared to the main simulation, with many locations experiencing deglaciation 200-600 yrs earlier than the standard simulation.

[Figure]

Figure S. A) The difference in the simulated deglaciation age between sensitivity experiment 2 and the main simulation. B.) The difference in the simulated deglaciation age between sensitivity experiment 3 and the main simulation. C.) The difference in the simulated deglaciation age between sensitivity experiment 4 and the main simulation. Blue colors indicate slower ice retreat for the sensitivity experiments compared to the main simulation, while red colors indicate faster ice retreat for the sensitivity experiments compared to the main run.

In Figure SB the difference in the deglaciation age between sensitivity experiment 3 the main simulation and is shown, where precipitation is held constant at the monthly mean 12.5 ka- 12 ka value. Winter precipitation during 12.5 ka – 12 ka is reduced modestly by up to 7% relative to the preindustrial across the model domain (see Figure 9 current manuscript). Ice retreats faster across most portions of the model domain, particularly along the ice margins through the interior. Deglaciation along the margins occurs >1 kyr in many locations, and between 200 yrs to 1 kyr faster across portions of the ice interior.

In Figure SC the difference in the deglaciation age between sensitivity experiment 4 and the main simulation is shown, where precipitation is held constant at the monthly mean 22 ka- 20 ka value. This simulation is discussed in our current manuscript (current Figure 7B), but here we show the difference from the main simulation. Winter precipitation during 22 ka – 20 ka is increased by up to 10% relative to the preindustrial (Figure 9 current manuscript) across the northern portion of the model domain (North of 40S), but is similar to preindustrial values across the southern portion of our model domain (South of 40S). As discussed in the current manuscript, with higher precipitation across the northern portion of the model domain, ice retreats slower during the last deglaciation relative to our main simulation by >1 kyr (and in some locations up to 2 kyr).

Much of the deglacial winter precipitation anomalies are within 10% of the preindustrial values (Figure 9). While temperature is the main driver of ice retreat (as shown by previous studies cited in the manuscript), these sensitivity studies suggest that modest changes in precipitation can alter the pace of ice retreat. While we have shown with the analysis of the TraCE-21ka outputs that the circulation changes and precipitation anomalies simulated across the last deglaciation qualitatively match well with paleoclimate proxies across the CLD (please see our discussion to RC 2A above and associated figures provided), many of those paleoclimate proxies indicate that the LGM and

early deglaciation may have been up to 2 times wetter than present day (Moreno et al., 1999; 2015). Therefore, because the anomalies of precipitations during the early deglaciation relative to the preindustrial as simulated by TraCE-21ka are smaller than those anomalies in precipitation found in the proxy records, we can deduct from our sensitivity analysis here that precipitation across the CLD, forced by proposed changes in the SWW (Moreno et al., 1999;2015) may have helped offset melt from deglacial warming thereby influencing the pacing of early deglacial ice retreat in this region.

*How we plan to revise this section._ Please note we refer to the new figure above as Figure S. We will give this the appropriate figure # in the updated manuscript:

We will plan to Revise Figure 7 and instead just show the resulting deglaciation age for the main simulation. Then in section 3.2.2 (Sensitivity Tests), we will add the panel figure above, which shows the results from the sensitivity experiments where precipitation is held fixed across the last deglaciation experiments. Text below will be added to section 3.2.2 as:

"To better assess how changes in precipitation may modulate the deglaciation across the CLD we perform additional sensitivity tests. We refer to the simulation discussed above as our *main run*, where the climate boundary conditions of temperature and precipitation varied temporally and spatially across the last deglaciation. Three more simulations are performed where temperature is allowed to vary across the last deglaciation, but precipitation remains fixed at a given magnitude for a particular time interval. Each experiment is listed below as:

1) Monthly precipitation is held constant at the preindustrial mean. Displayed in Figure 9, wintertime preindustrial precipitation reduced compared to the period 22 ka to 18 ka, but is higher than what is simulated after 18 ka for the exception of the ACR at 14.5 ka.
2) Monthly precipitation is held constant at the mean 12.5 ka-12 ka values. Displayed in Figure 9 this is a period of reduced precipitation relative to the preindustrial (~7% reduction).
3) Monthly precipitation is held constant to the 22-20 ka mean, which is roughly 10% higher than preindustrial values across the Northern portion of the model domain (North of 40°S).

In Figure SA, the difference in the deglaciation age between sensitivity experiment 1 and the main simulation is shown, where precipitation is held constant at the preindustrial value. Across our model domain, wintertime precipitation during the preindustrial is reduced compared to the early deglaciation (22 ka to 18ka) and is similar to slightly higher particularly south of 40°S after 18 ka (Figure 9). When holding precipitation constant at the preindustrial value through the last deglaciation, we notice that ice retreats faster across most portions of the model domain, particularly along the ice margins and in area north of 40°S. In the southern portion of our model domain (south of 40°S), where the changes in deglacial precipitation relative to the preindustrial are lower (Figure 9), the difference in simulated deglaciation age are also smaller. In general, the pace of deglaciation increases by up to 1 kyr compared to the main simulation, with many locations experiencing deglaciation 200-600 yrs earlier than the main simulation.

In Figure SB the difference in the deglaciation age between sensitivity experiment 2 and the main simulation is shown, where precipitation is held constant at the monthly mean 12.5 ka- 12 ka value.

Winter precipitation during 12.5 ka – 12 ka is reduced modestly by up to 7% relative to the preindustrial across the model domain (Figure 9). Ice retreats faster across most portions of the model domain, along the ice margins through the interior. Deglaciation along the margins occurs >1 kyr in many locations, and between 200 yrs to 1 kyr faster across portions of the ice interior.

In Figure SC the difference in the deglaciation age between sensitivity experiment 3 and the main simulation is shown, where precipitation is held constant at the monthly mean 22 ka- 20 ka value. Winter precipitation during 22 ka – 20 ka is increased by up to 10% relative to the preindustrial (Figure 9) across the northern portion of the model domain (north of 40°S), but is similar to preindustrial values across the southern portion of our model domain (south of 40°S). With the imposed higher precipitation across the northern portion of the model domain, ice retreats slower during the last deglaciation relative to our standard simulation by >1 kyr, and in some locations up to 2 kyr."

We also will plan to add text to section 4.1 Ice-climate sensitivity, while removing the text concerning the PMIP4 simulations. This is in addition to the added text described in RC 2a above:

"Much of the TraCE-21ka simulated winter precipitation anomalies shown in Figure 9 are within 10% of the preindustrial value. The sensitivity tests conducted here suggest that modest changes (~10%) in precipitation can alter the pace of ice retreat across the CLD on timescales consistent with the resolution of geochronological proxies constraining past ice retreat. We note that while TraCE-21ka simulates variations in the precipitation across our model domain that are consistent with hydroclimate proxies discussed above (Moreno et al., 1999; 2015; 2018), the changes are not as large as proxy data across the CLD indicate. For example, hydroclimate proxies suggest that the LGM and early deglaciation was up to 2 times wetter across the CLD than present day (Moreno et al., 1999; Heusser et al., 1999). Therefore, we can deduct from our sensitivity analysis here that higher precipitation anomalies during the LGM and last deglaciation, forced by proposed changes in the SWW (Moreno et al., 1999;2015), may have helped offset melt from deglacial warming thereby influencing the pacing of early deglacial ice retreat in this region."

- Specific suggestions:

    - Figure 6 takes too much space for the content it presents.

        This has been removed as we have taken out the section using PMIP models.

- Line 428 & similar instances: Provide coordinates of these locations. Most people would not know where to look for them.

We have added locations to the Map (figure 2).

[Figure]

Figure 1. Bedrock topography for our study area (meters). Our model domain (shown as the black line), encompasses the reconstructed LGM ice limit (shown in red) from PATICE (Davies et al., 2020). Present day lakes are shown in blue, with abbreviated names as: SR (Seno de Reloncaví), GA (Golfo de Ancud), LL (**Lago Llanquihue), LR1 (Lago Rupanco)**, LP1 (Lago Puyehue), LR2 (Lago Ranco), LR3 (Lago Riñihue), LP2 (Lago Panguipulli), LC (Lago Calafquén), LV (Lago Villarica), LNH (Lago Nahuel Huapi).

- Figure 11: Missing the ice area from PATICE for 10 ka?
  We have added the area from the PATICE reconstruction for the 10 ka interval to the plot (figure 11).

[Figure]

Figure 2.  The simulated ice area (km²) from 22 ka to 10 ka shown as the black line.  The red dots indicate the calculated ice area across our model domain for the reconstructed ice extent from PATICE (Davies et al., 2020).

- Conclusions present some departures from the statistics found in the results and discussion. Could these be refined?

We have updated the Conclusions below:

"In this study, we use a numerical ice sheet model to simulate the LGM and deglacial ice history across the northernmost extent of the PIS, the CLD.  The ice sheet model relied on inputs of temperature and precipitation from the TraCE-21ka climate model simulation covering the last 22,000 years in order to simulate the deglaciation of the PIS across the CLD into the early Holocene.

Our numerical simulation suggests that large scale ice retreat occurs after 19 ka coincident with rapid deglacial warming, with the northern portion of the CLD becoming ice free by 17 ka.  The simulated ice retreat agrees well with the most comprehensive geologic assessment of past PIS history available (PATICE; Davies et al., 2020) for the LGM ice extent and early deglacial but diverge when considering the ice geometry at and after 15 ka.  In our simulations, the PIS persists

until 15 ka across the remainder of the CLD, followed by ice retreat to higher elevations as mountain glaciers and small ice caps persist into the early Holocene (e.g., Cerro Tronador). The geologic reconstruction from PATICE instead estimates a small ice cap persisting across the southern portion of high terrain in the CLD until about 10 ka. However, because there are limited geologic constraints particularly after 15 ka, high uncertainty in the timing and extent of deglacial ice history remains in the geologic reconstruction. Therefore, our results provide an additional reconstruction of the deglaciation of the PIS across the CLD that differs from PATICE after 15 ka, emphasizing a need for future work that aims to improve geologic reconstructions of past ice margin migration particularly during the later deglaciation across this region.

While deglacial warming was a primary driver of the demise of the PIS across the last deglaciation, we find that precipitation modulates the pacing and magnitude of deglacial ice retreat across the CLD. Paleoclimate proxies within the CLD has shown that the strength and position of the SWW varied during the LGM and last deglaciation, altering hydrologic patterns and influencing the deglacial mass balance. We find that the simulated changes in the strength and position of the SWW in TraCE-21ka are similar to those inferred by paleoclimate proxies of precipitation, consistent with a wetter than preindustrial climate being simulated and reconstructed over the CLD and in particular the region north of 40°S. Through a series of sensitivity tests, we alter the magnitude of the precipitation anomaly modestly (up to 10%) during our transient deglacial simulations and find that the pacing of ice retreat can speed up or slow down by a few hundred years and up to 2000 years depending on whether we impose an increase or decrease in the precipitation anomaly. While paleoclimate proxies of precipitation suggest that the CLD may have experienced twice as much precipitation during the LGM and early deglacial relative to present day (Moreno et al., 1999;2015), TraCE-21ka simulates smaller increases in LGM and early deglacial precipitation (~10-15% greater than preindustrial). Therefore, while our modelling suggests that modest changes in precipitation can modulate the pace of deglacial ice retreat across the CLD, from our analysis we can deduct that larger anomalies in precipitation as found in the paleoclimate proxies may have an even larger impact on modulating deglacial ice retreat. Because paleoclimate proxies of past precipitation are often lacking, and climate models can simulate a range of possible LGM and deglacial hydrologic states, these results suggest that improved knowledge of the past precipitation is critical towards better understanding the drivers of PIS growth and demise, especially as small variations in precipitation can modulate ice sheet history on scales consistent with geologic proxies."

**JJA 925 hPA zonal wind**

**Figure S3 A-E.** First column: The difference in the JJA 925 hPa zonal wind for each corresponding time period relative to the PI (in m/s). Positive values indicate increased zonal wind speed and negative values indicate decreased zonal wind speed relative to the PI. The magenta line is the position of the maximum zonal wind during the PI. The black line is the position of the maximum zonal wind for the corresponding time period. The red polygon denotes the location of our model domain. Second column: Zonal mean JJA 925 hPa wind (in m/s) averaged over -85 to -55 degrees west longitude for the PI and the corresponding time period. The time periods listed are computed over 500 yr periods (*LGM*: 22ka-21.5ka; *18ka*: 18.5ka-18ka; *16ka*: 16.5ka-16ka;

[Figure]

[Figure]

**850 hPa Moisture Flux Convergence**

We calculate the moisture flux convergence (MFC) at 850 hPa as:

$$-1(\nabla \cdot Vq)$$

where $\nabla$ is the gradient operator, V is the horizontal wind vector (u,v), and q is the specific humidity. The divergence is multiplied in this case by -1 to show moisture flux convergence. We calculate the MFC at the **LGM** (22ka-21.5ka), ***18ka*** (18.5ka-18ka), ***16ka*** (16.5ka-16ka), ***14ka*** (14.5ka-14ka), and ***12ka*** (12.5ka-12ka) and then compute the difference at these periods against the **PI** (shown below in Figure S4A-E).

**Figure S4 A-E.** JJA moisture flux convergence at 850 hPA for each corresponding time period, computed as the difference from the PI (in g kg$^{-1}$s$^{-1}$). Positive anomalies indicate areas of greater moisture flux convergence relative to the PI and negative values indicate areas of greater moisture flux divergence relative to the PI. The wind vectors correspond to the anomaly relative to the PI for the 850 hPa pressure level. The red polygon denotes the location of our model domain. The time periods listed are computed over 500 yr periods (*LGM*: 22ka-21.5ka; *18ka*: 18.5ka-18ka; *16ka*: 16.5ka-16ka; *14ka*: 14.5ka-14ka; *12ka*: 12.5ka-12ka).

[Figure]

[Figure]

---

## Referee Report (RR1)

**Review of Cuzzone et al. 'Modeling the timing of Patagonian Ice Sheet retreat in the Chilean Lake District from 22-10 ka', *The Cryosphere***

**Paper Summary:**

Cuzzone et al. present a numerical ice sheet model reconstruction of the Patagonian Ice Sheet in the region of the Chilean Lake District (CLD), focusing on the LGM and subsequent deglaciation. The CLD is an area that is generally lacking empirical data and this work represents a meaningful contribution to our understanding of the drivers of ice evolution following the LGM. The authors use the model ISSM to simulate Patagonian Ice Sheet evolution in the region which they combine with the TraCE-21ka climate simulation. They use this to: 1) provide constraints on the nature of CLD ice retreat and 2) assess the possible key climatic controls on ice evolution during deglaciation. They provide a detailed methodology and the results of their experiments are clear, highlighting the role of precipitation in modulating ice recession during deglaciation. The authors highlight uncertainties in the model and justify their choices in model setup and design (e.g., GIA, calving).

The authors have been very comprehensive in addressing previous comments on the manuscript. I only have minor comments on the resubmitted manuscript with reference to the methods and results, which are clear from my viewpoint. The discussion in relation to climate is also clear. The majority of weaknesses that remain relate to the discussion, where the descriptions, interpretations, and comparisons to empirical data need to be re-phrased or clarified- some points just do not make sense (e.g., discussion and limitations). I have provided directions for the authors in relation to those. Once these minor corrections are complete (which shouldn't take the authors long at all) I recommend this work, which represents a substantial contribution to our understanding of glacier-climate interactions in the CLD, be published.

**Comments on the Manuscript**

**Introduction:**

**Line 51:** '…across an area presently known as the Chilean Lake District'- add latitude and longitude for CLD area here. This is helpful context, particularly as you are later comparing to the SWW.

**Lines 57/58:** + and outwash plains, though I think a lot of constraints in this region are also from sedimentological work on exposures, whereas constraints in the east of the CLD are extremely lacking. Maybe add '**only**… well constrained'.

**Line 59/60"** I had to read this a few times as you have just said the LGM limits are well constrained in the SW/W, maybe add: '… due to a lack of geomorphological and geochronological constraints on ice-margin change **following the LGM,** the reconstructed deglaciation remains highly uncertain'.

**Line 63/65:** Maybe best to throw a line explaining the Southern Annular mode in this paragraph here? Not everyone will be familiar with regional + SH climate(drivers).

**Methods/Results:**

The methods and results of this work have been outlined very clearly and resolved in previous iterations, and are combined with a very detailed supplemental document. Just a few clarifications here:

**Line 234:** 'We simulate calving where the PIS interacts with ocean.' -but not with lakes, add brief sentence so this is clear early on?- I know this is justified later, but here I wondered why it wasn't included given substantial numbers of lakes across the 'Chilean Lake District'.

**Lines 330-333:** What do you mean by deglaciation? Significant re-advances would imply a punctuated net retreat, versus small glacier fluctuations we would expect to see on an annual basis as a result of seasonal differences in mass balance? Maybe just define what you mean by deglaciation here, as interpretations do vary.

**Discussion:**

**Lines 440-441:** '…as limitations in paleo-proxy data and disagreement between climate models prohibit certainty'- '…due to disagreement between paleo-proxy data and climate models'? Or something like this, I struggled to follow.

**Lines 443-444:** '…linking the paleoclimate change in SWW position and strength from regional paleoclimate proxies remains problematic (Kohfeld et al., 2013).'- again, this was hard to read. What do you mean, re-phrase?

The comparison to climate is detailed, but the discussion of ice retreat and the limitations section can definitely be built upon:

**Line 552:** Particularly in the CLD- as you have highlighted?

**Lines 559-560:** Across the CLD, the LGM ice extent is well constrained by geologic proxies particularly in the west and southwest- clarify the timeframe from the panels in Figure 1.

**Lines 560-563:** 'The moraines that constrain the piedmont ice lobes that formed along the western boundary are now presently lakes and have reasonable age control (Denton et al., 1999; Moreno et al., 1999; Lowell et al., 1995), giving confidence to the LGM ice margin limits.'- How can moraines be lakes? I know what you are trying to say, but this needs re-writing. These north-western outlet glaciers deposited moraines that lay west of the overdeepenings that are occupied by modern lakes. Clarify.

**Lines 571-572:** '…cosmogenic nuclide surface exposure date retrieved from the Nahel Huapi moraine yielded an age of ~31.4 ka (Zech et al., 2017).'- Where is this moraine? Is it on a figure? We need some idea where it is, even coordinates or a dot on a map somewhere.

**Lines 578-581:** 'In regards to ice area and extent, our simulated ice sheet at the LGM using TraCE-21ka climate boundary conditions agrees well with the PATICE reconstruction (Figure 10). Our simulations reveal that deglaciation began between 19 ka to 18 ka, consistent with

the geologic proxies (Davies et al., 2020)'.- Can we add a line or two of comparison down the PIS as this is your discussion? This is similar to the Lago Palena/General Vintter (Soteres et al., 2022), Corcovado (Leger et al., 2021, QSR) and Cisnes (Garcia et al, 2019) glaciers southeast of the CLD (~43-44 deg S). Is it different to what is going on even further South (LGC/BA or Tierra Del Fuego)? Or on Isla Chiloe? Worth a mention at least of similarities and differences along the transect of the Andes here. (Note that this is a different Leger paper to the one you mention, same year).

**Lines 584-585:** 'After 15 ka, mountain glaciers remain in our simulation but there is no presence of a large ice cap as reconstructed in PATICE'- there is no data really at all for this time slice in PATICE. I think you can highlight this at various points here, more data is needed in the CLD to better evaluate this and future models. You have done what you can to evaluate this with available empirical data.

**Line 588:** '…region largely retreated by 15 ka, with only mountain glaciers remaining.' There is support for this further south. E.g., ice had receded enough for Atlantic-Pacific drainage routes to open east (and drainage reversed) by at around ~16.3 ka (e.g., Leger et al., 2021; QSR; ~44 Deg S)- which supports that the ice sheet had begun to unzip back into smaller ice caps as you show here. Would this not also support the same occurring further north (or even earlier possibly) in the CLD?

**Lines 594-595:** 'This potential for a favourable and prolonged period of glacier growth is likely missing in our simulations during the ACR, which may explain some of the mismatch against the PATICE reconstruction at 15 ka – 13 ka'. I realise you have a short ACR, but there are not constrains on these time slices in PATICE. You can probably remove or re-phrase this, and highlight that the mismatch is hard to assess due to data paucity here. This is generally a Patagonia-wide issue (we need more constraints).

**Limitations section of Discussion:**

There are a few points when referring to the geomorphology that this section is a little muddled-here are a few (minor) points to clarify:

**Line 616:** Re. evidence- are the moraines not comprised of glacio-tectonised outwash (check paper)? Be specific with the evidence. Just provide the details mentioned in the paper here. 'Moraines formed of glacio-tectonised outwash provide evidence for…'. This is helpful context (and it will keep the geomorphologists happy).

**Lines 617-619:** 'The formation of ice-contact proglacial lakes likely occurred as a function of deglacial warming and ice retreat (Bentley, 1996)'. Well, kind of, but really it is due to mass loss combined with the underlying bed topography (restricting proglacial drainage). Not all glaciers have lakes, it is a product of retreating into their overdeepenings where meltwater becomes trapped. Re-phrase? E.g., examples outlined in recent papers by Dave Evans (Iceland).

**Lines 620-624:** This doesn't really make sense: **'**During deglaciation, iceberg calving into the proglacial lakes may have occurred (Bentley 1996,1997; Davies et al., 2020), with evidence suggesting that local topography and calving may have controlled the spatially irregular timing of abandonment from the terminal moraines surrounding the proglacial lakes (Bentley, 1997).

1) If you have an ice-contact lake, you have calving in the ablation area… re. '…iceberg calving into the proglacial lakes may have occurred'. This is not necessary; remove or re-phrase point.

2) Surely calving cannot control the glacier abandonment of the terminal moraine positions surrounding the proglacial lakes, because the glaciers had not yet developed lake? Am I missing something? Please clarify/re-write this point so that it is clear.

**Lines 627-629:** 'However, across our region, Heirman et al. (2011) indicate that it is not well constrained how the proglacial lakes in the CLD may have influenced local deglaciation, as more geomorphic data is needed'. It's true that we do not have geomorphological data, but a lot is probably at the base of these lakes (if at all) and would require bathymetric surveys. There has been some work done on glaciolacustrine varve sediments around Lago Buenos Aires (see: Bendle et al., 2017/2019 (?) papers) supports a period of rapid recession following the onset of calving as glaciers retreated and formed lakes in their overdeepenings. Maybe reference some of this work? It builds on these points.

**Conclusions:**

- If you add these additional comparisons to the discussion, can we feed a line or two in here summarising the key points?

**Figures:**

**Figure 1:** Generally, I would make sure that maps have a north arrow, scale bar, and grid (latitude/longitude). Not necessary for all panels, but at least one to provide context.

**Figure 5:** I am struggling to follow this description. Is it not dark blue where ice is persisting after 10 ka? Is something wrong here either in the caption or the figure? Please clarify and fix if needed.

I hope that my comments are of some use to you and I look forward to seeing this published in the near future.

---

## Author Response (AR2)

**Response in Red**

**Report #1:**

The authors present a modelling reconstruction of the northern branch of the Patagonian Ice Sheet (PIS) during the last deglaciation, precisely from the last glacial maximum (LGM) to 10 ka ago. The ice-flow model ISSM is used for this purpose. The exercise makes use of a transient climatology for the last deglaciation (TraCE-21ka) to reconstruct both the glacial state of the ice sheet and to simulate the early stage of the deglaciation. The results are then compared with reconstructions available for that region (PATICE - Davies et al., 2020). Ice-climate interactions, sensitivity on the employed climate model, and goodness of the model results are then discussed.

I would like to extremely thank the authors for the exhaustive response to the criticisms that I and Reviewer#2 raised in the first round of reviews. I see there has been a huge effort in terms of clarification of the methodology and corroboration of the conclusions, improved by additional model experiments to investigate the thermal state of the PIS and the response of the ice sheet to precipitation changes throughout the last deglaciation. Also, I would like to thank the authors for the clarification regarding the GIA forcing prescribed in their simulations, which was one of my major concerns. I think that the paragraphs added in the "Limitations" section and that in the supplementary material clarify well this aspect.

The results are sound and the manuscript is well written. I therefore suggest the publication of their manuscript in TC after minor corrections, that I note down here:

We would like to thank the reviewer again for their thorough assessment of our work and especially thank the reviewer for agreeing to reassess the changes made to the original draft. We feel that the reviewer's comments and edits made following their recommendations have greatly improved our manuscript.

Prescribed GIA
The authors claim that "the prescribed GIA is in good agreement (Figure S2) with a reconstruction of relative sea level change from an insolation basin in central Patagonia". I believe the authors but I don't see any comparison in Figure S2. Could you please show this agreement by comparing the two RSL curves?

Certainly. We have updated our supplemental Figure S2. There we added notation denoting the total uplift reconstructed by Troch et al., 2022 as well as the rates of uplift reconstructed by Troch et al., 2022 between 16.5 ka and 9 ka and 9 ka and 0ka, and compared this to the prescribed rates of uplift in our study.

LGM climate description
I don't see the point of describing the LGM temperature and precipitation anomalies from TraCE21-ka in section 3.1 of the results, since these are not an outcome of the ice-flow modelling simulation, rather an output from CCSM3. I would rather shift Figure 3 and its description in the main text to section 2.4, where the climatic forcings are described. Also, it would be interesting to see the anomalies for other time snapshots during the deglaciation (e.g. 17 ka, 15 ka, 12 ka). By doing that, the description of the climate at the LGM would gain more interest as compared to other climates throughout the last 20 ka and it would add a spatial information to Figure 7 for the sensitivity tests.

Thanks. We have updated Figure 3 by moving it up to section 2.4 and have added anomaly maps for 17 ka, 15 ka, and 12 ka. We updated some text to reflect these changes in section 2.4 and the figure caption, as well as text in the section 3.2.2 and 4.1.

Sensitivity tests
I think it would be easier for the reader that the name of experiments 1, 2 and 3 directly be called the precipitation change that is applied in the sensitivity test. I would suggest to name these experiments as "prec_PI" (instead of experiment 1), "prec_12ka" and "prec_LGM", or something alike.

We agree. We have changed the experiment names to: Precip. PI, Precip. 12 ka, and Precip. LGM.

Mathematical expressions

This is a technical comment, but please, be careful on how you write the equations in the Methods section. Many vectors miss the proper notation (e.g. basal shear stress and basal velocity in eq. 1; frontal velocity, horizontal velocity and normal vector pointing out from the calving front in eq. 5; horizontal velocity in eq. 6).

Done

Specific comments:

P7 L206-209. I would also cite Figure 4 to give a spatial overview of this agreement

Done.

P7 L235. I believe the reference for the level set method should be Choi et al., 2021, equation 3 (Equation 3 from Bondzio et al., 2016 is the ice thickness equation)

Thanks for double checking. Consulting with other ISSM team members, the levelset method is indeed introduced in Bondzio et al., 2016, so we prefer to stick with that citation.

P7 L245 and P8 L251. Please change tensile "strength" to tensile "stress".

Done

P7 L246-252. I would rephrase that as "… is the maximum stress threshold which has separate values for tidewater and floating ice, namely 1 MPa and 200 kPa respectively. The ice front will retreat if the von Mises tensile stress exceeds the user defined stress threshold.". Also, is it really needed to specify the threshold for floating ice? I don't think that ice shelves are modelled in your domain.

Thanks for the suggestion. We have updated the text accordingly, but kept the floating ice stress threshold in there for completeness since it was set in the model runs despite no to limited floating extensions forming.

Figure 3. Please add "The bilinearly interpolated LGM summer (DJF)…" to the caption.

Done.

P8 L267-274. As I mentioned in the general comments, I would shift the LGM climate description to the section about Climate Forcings.

Done.

P9 L274-278. If I understand correctly the position of SWW simulated in TracE-21ka at the LGM is in contrast to what is found in Kohfeld et al., 2013 and Moreno et al., 2015, who simulate a glacial equatorial shift of the SWW. I think you describe well this discrepancy in P17

Thank you. Yes, you are correct and we address this in the discussion.

L 508-527, but could you add a sentence about this also in the introduction when you comment about latitudinal changes in SWW? You could also remove P9 L 274-278, since you tackle this point well in the discussion.

We are not 100% certain if the reviewer is asking us to note in the introduction the relatively good agreement between TraCE-21ka and paleoproxies across the CLD. If so, we would like to leave the introduction text as is, as to our knowledge, our manuscript is the first to directly compare TraCE-21ka qualitatively against paleoproxies of precipitation in the CLD. Regarding L274-278. That is a good point and we have removed this text.

P9 L283. Please add "(Figure 4)" at the end of the sentence.

Done

P9 L288. Please add "(Figure 10)" at the end of the sentence.

Done

Figure 7. Could you add a shadow area to highlight the time intervals at 12-12.5 ka and between 22 and 20 ka?

Done. And we added text in figure caption: "Intervals of time used in the sensitivity test are highlighted by gray shading."

P12 L379. I believe for experiment 2 you mean experiment 1, but I would rather suggest to name this experiment as "prec_PI" or something alike.

Yes, thank you. We have corrected this and used your recommended naming.

Figure 8. Experiments 2, 3 and 4 should be 1, 2 and 3, or "prec_PI", "prec_12ka" and "prec_LGM".

Done.

P13 L390-398. I would restructure this paragraph so that you first describe experiment 2 (or "prec_12ka") and then experiment 3 (or "prec_LGM").

We have restructured following your recommendation. The text now reads: "For experiment *Precip. 12 ka*, winter precipitation is reduced by up to 7% (Figure 8B) relative to the preindustrial across the model domain (Figure 3 and 7). In this experiment ice retreats faster across most of the CLD, from the ice margins and through the interior. Deglaciation along the margins occurs >1 kyr faster in many locations, and between 200 yrs to 1 kyr faster across portions of the ice interior. For experiment *Precip LGM*, winter precipitation is increased by up to 10% (Figure 8C; *Precip LGM*:) across the northern portion of the model domain (north of 40°S) relative to preindustrial, but is similar to preindustrial values across the southern portion of our model domain (south of 40°S). In this experiment, with the imposed higher precipitation across the northern portion of the model domain, ice retreats slower during the last deglaciation relative to our standard simulation by >1 kyr, and in some locations up to 2 kyr."

Figure 9. Could you choose another colour for the simulated ice extent at 15.1 ka (e.g. orange, yellow)?

We prefer to keep the current colormap. We tried many combinations and this seemed to be the best visually.

Figure 10. Could you please add a line at 15.7 ka or a symbol to facilitate the comparison with the PATICE reconstruction at 15 ka?

We have added a green rectangle to highlight the simulated ice area at 15.7 ka.

**Report #2:**

Paper Summary:
Cuzzone et al. present a numerical ice sheet model reconstruction of the Patagonian Ice Sheet in the region of the Chilean Lake District (CLD), focusing on the LGM and subsequent deglaciation. The CLD is an area that is generally lacking empirical data and this work represents a meaningful contribution to our understanding of the drivers of ice evolution following the LGM. The authors use the model ISSM to simulate Patagonian Ice Sheet evolution in the region which they combine with the TraCE-21ka climate simulation. They use this to: 1) provide constraints on the nature of CLD ice retreat and 2) assess the possible key climatic controls on ice evolution during deglaciation. They provide a detailed methodology and the results of their experiments are clear, highlighting the role of precipitation in modulating ice recession during deglaciation. The authors highlight uncertainties in the model and justify their choices in model setup and design (e.g., GIA, calving).

The authors have been very comprehensive in addressing previous comments on the manuscript. I only have minor comments on the resubmitted manuscript with reference to the methods and results, which are clear from my viewpoint. The discussion in relation to climate is also clear. The majority of weaknesses that remain relate to the discussion, where the descriptions, interpretations, and comparisons to empirical data need to be re-phrased or clarified- some points just do not make sense (e.g., discussion of ice retreat and limitations). I have provided directions for the authors in relation to those. Once these minor corrections are complete (which shouldn't take the authors long at all) I recommend this work, which represents a substantial contribution to our understanding of glacier-climate interactions in the CLD, be published.

We greatly appreciate the reviewer for their thorough remarks and comments, support for our work, and lastly for stepping in to review this 2nd submission. And thank you for clarifying some of our misunderstandings with the regional geomorphology.

Comments on the Manuscript
Introduction:
Line 51: '…across an area presently known as the Chilean Lake District'- add latitude and longitude for CLD area here. This is helpful context, particularly as you are later comparing to the SWW.

Done.

Lines 57/58: + and outwash plains, though I think a lot of constraints in this region are also from sedimentological work on exposures, whereas constraints in the east of the CLD are extremely lacking. Maybe add 'only… well constrained'.

Done.

Line 59/60" I had to read this a few times as you have just said the LGM limits are well constrained in the SW/W, maybe add: '… due to a lack of geomorphological and geochronological constraints on ice-margin change following the LGM, the reconstructed deglaciation remains highly uncertain'.

Done.

Line 63/65: Maybe best to throw a line explaining the Southern Annular mode in this paragraph here? Not everyone will be familiar with regional + SH climate(drivers).

Certainly. We have added some text here: "The wintertime climate across South America is strongly influenced by the southern annular mode (SAM; Hartmann and Lo, 1998), by which its phase and strength is regulated by changes in the difference of zonal mean sea-level pressure between mid (40°S) and high latitudes (65°S). The SAM in turn modulates the strength and position of the southern westerly winds (SWW) over decadal to multi-centennial timescales, which exert a large control on the synoptic scale hydrologic and heat budget (Garreaud et al., 2013)."

Methods/Results:
The methods and results of this work have been outlined very clearly and resolved in previous iterations, and are combined with a very detailed supplemental document. Just a few clarifications here:
Line 234: 'We simulate calving where the PIS interacts with ocean.' -but not with lakes, add brief sentence so this is clear early on?- I know this is justified later, but here I wondered why it wasn't included given substantial numbers of lakes across the 'Chilean Lake District'.

We have modified the text here: "We simulate calving where the PIS interacts with ocean, but do not include any treatment of calving in proglacial lakes (see section 4.3)."

We recognize that it would be nice to have included some treatment of calving in proglacial lakes. However, as discussed in prior response to reviewers and in section 4.3, this methodology within ice sheet models is still novel and not well established. It is the subject of some current work with ISSM.

Lines 330-333: What do you mean by deglaciation? Significant re-advances would imply a punctuated net retreat, versus small glacier fluctuations we would expect to see on an annual basis as a result of seasonal differences in mass balance? Maybe just define what you mean by deglaciation here, as interpretations do vary.

Currently in the text, we define deglaciation as the point in time when the grid cell becomes ice free. However, from that definition and due to possibility of readvance, a particular grid point could experience deglaciation multiple times (readvance/retreat). Therefore, we take the youngest age of when a grid cell deglaciates (or ice retreats from). We have modified the text as: "From the resulting transient simulation, we calculate the timing of deglaciation across our model domain (Figure 5) as the youngest age at which grid points become ice free."

Discussion:
Lines 440-441: '…as limitations in paleo-proxy data and disagreement between climate models prohibit certainty'-'…due to disagreement between paleo-proxy data and climate models'? Or something like this, I struggled to follow.

We have clarified this as: "Determining the influence of the SWW on the heat and hydrologic budget across South America during the LGM and last deglaciation remains difficult, as paleo-proxy data is limited and climate models tend to disagree on the evolution of the SWW (Kohfeld, 2013; Berman et al., 2018)."

Lines 443-444: '…linking the paleoclimate change in SWW position and strength from regional paleoclimate proxies remains problematic (Kohfeld et al., 2013).'- again, this was hard to read. What do you mean, re-phrase?

We have changed this sentence to: "And while paleo-proxy evidence does suggest wetter conditions across the CLD during the late glacial (Moreno and Videla, 2018), linking this variability to changes in the position and strength of the SWW remains difficult (Kohfeld et al., 2013)."

The comparison to climate is detailed, but the discussion of ice retreat and the limitations section can definitely be built upon:
Line 552: Particularly in the CLD- as you have highlighted?

Yes. We have adjusted:" Because geochronological constraints on past PIS change are limited, particularly in the CLD, the PATICE reconstruction assigns qualitative confidence to its reconstructed ice margins."

Lines 559-560: Across the CLD, the LGM ice extent is well constrained by geologic proxies particularly in the west and southwest- clarify the timeframe from the panels in Figure 1.

We have adjusted the text: "Across the CLD, the LGM (25 ka, 20 ka) ice extent is well constrained by geologic proxies particularly in the west and southwest (Figure 1)."

Lines 560-563: 'The moraines that constrain the piedmont ice lobes that formed along the western boundary are now presently lakes and have reasonable age control (Denton et al., 1999; Moreno et al., 1999; Lowell et al., 1995), giving confidence to the LGM ice margin limits.'- How can moraines be lakes? I know what you are trying to say, but this needs re-writing. These north-western outlet glaciers deposited moraines that lay west of the overdeepenings that are occupied by modern lakes. Clarify.

We have clarified by adjusting the text: "The moraines that constrain the piedmont ice lobes that formed along the western boundary have reasonable age control (Denton et al., 1999; Moreno et al., 1999; Lowell et al., 1995), giving confidence to the LGM ice margin limits."

Lines 571-572: '…cosmogenic nuclide surface exposure date retrieved from the Nahel Huapi moraine yielded an age of ~31.4 ka (Zech et al., 2017).'- Where is this moraine? Is it on a figure? We need some idea where it is, even coordinates or a dot on a map somewhere.

We have updated the text here with some coordinates:

"For instance, a single cosmogenic nuclide surface exposure date retrieved from the Nahuel Huapi moraine yielded an age of ~31.4 ka (Zech et al., 2017; 41.04° S, 71.15° W)."

Lines 578-581: 'In regards to ice area and extent, our simulated ice sheet at the LGM using TraCE-21ka climate boundary conditions agrees well with the PATICE reconstruction (Figure 10). Our simulations reveal that deglaciation began between 19 ka to 18 ka, consistent with the geologic proxies (Davies et al., 2020)'.- Can we add a line or two of comparison down the PIS as this is your discussion? This is similar to the Lago Palena/General Vintter (Soteres et al., 2022), Corcovado (Leger et al., 2021, QSR) and Cisnes (Garcia et al, 2019) glaciers southeast of the CLD (~43-44 deg S). Is it different to what is going on even further South (LGC/BA or Tierra Del Fuego)? Or on Isla Chiloe? Worth a mention at least of similarities and differences along the transect of the Andes here. (Note that this is a different Leger paper to the one you mention, same year).

We have updated the text to reflect the recommendations of the reviewer:

"In regards to ice area and extent, our simulated ice sheet at the LGM using TraCE-21ka climate boundary conditions agrees well with the PATICE reconstruction (Figure 10). Our simulations reveal that deglaciation began between 19 ka to 18 ka, consistent with the Davies et al. (2020) reconstruction. Notably, the simulated timing of deglaciation agrees with moraine records further south on the eastern side, such as in Río Corcovado (~43° S, Leger et al., 2021a), Río Cisnes (~44° S, Garcia et al., 2019), Lago Palena/General Vintter (~44° S, Soteres et al., 2022), and Río Ñirehuao (~45° S, Peltier et al., 2023). On the other hand, glaciers are thought to have withdrawn from their LGM position later between ~18 - 17 ka on the northwestern margin (~41° S, Denton et al., 1999; Moreno et al., 2015), in the southern (~46° S, Kaplan et al., 2004), and southernmost regions (~52° S, McCulloch et al., 2000; 2005; Kaplan et al., 2008; Peltier et al., 2021). The simulated ice retreat continues until 15 ka, with the largest pulses in ice mass loss occurring at 18.6 ka, 16.8 ka, and 16 ka (Figure 6)."

Lines 584-585: 'After 15 ka, mountain glaciers remain in our simulation but there is no presence of a large ice cap as reconstructed in PATICE'- there is no data really at all for this time slice in PATICE. I think you can highlight this at various points here, more data is needed in the CLD to better evaluate this and future models. You have done what you can to evaluate this with available empirical data.

We agree, and have made attempts throughout the manuscript to inform the reader of the lack of data. The initial reviewers question why the model diverged from PATICE, but did not necessarily realize (from the text in the first submission), that PATICE is very uncertain and therefore is not a reliable constraint later in deglaciation. Here we add: "Comparison between the model simulations and PATICE becomes difficult during the 15 -13 ka period as confidence in the geologic reconstruction is low due to a lack of geochronological and geomorphological constraints on past ice history."

Line 588: '…region largely retreated by 15 ka, with only mountain glaciers remaining.' There is support for this further south. E.g., ice had receded enough for Atlantic-Pacific drainage routes to open east (and drainage reversed) by at around ~16.3 ka (e.g., Leger et al., 2021; QSR; ~44 Deg S)- which supports that the ice sheet had begun to unzip back into smaller ice caps as you show here. Would this not also support the same occurring further north (or even earlier possibly) in the CLD?

We have updated the text here: "This is supported further south, where the ice sheet disintegrated at ~16 ka with paleolake draining to the Pacific Ocean (~43° S, Leger et al., 2021a) and the ice remaining limited to higher mountain areas"

Lines 594-595: 'This potential for a favourable and prolonged period of glacier growth is likely missing in our simulations during the ACR, which may explain some of the mismatch against the PATICE reconstruction at 15 ka – 13 ka'. I realise you have a short ACR, but there are not constrains on these time slices in PATICE. You can probably remove or re-phrase this, and highlight that the mismatch is hard to assess due to data paucity here. This is generally a Patagonia-wide issue (we need more constraints).

We agree, but added these statements to address the comments of reviewers during the first submission. Since we address the scarcity of data in PATICE above these lines, we decided to change the text to: "This potential for a favorable and prolonged period of glacier growth is likely missing in our simulations during the ACR."

In those regards we acknowledge that the simulations may miss a window of potentially favorable glacier growth (which we think readers may find interesting), without again comparing our results during this interval to PATICE which is uncertain.

Limitations section of Discussion:
There are a few points when referring to the geomorphology that this section is a little muddled- here are a few (minor) points to clarify:
Line 616: Re. evidence- are the moraines not comprised of glacio-tectonised outwash (check paper)? Be specific with the evidence. Just provide the details mentioned in the paper here. 'Moraines formed of glacio-tectonised outwash provide evidence for…'. This is helpful context (and it will keep the geomorphologists happy).

Yes, Bentley 1996 describe glacio-tectonized outwash. We have adjusted the text: "Across most of our domain, moraines formed of glacio-tectonized outwash (Bentley, 1996) provide evidence for an advance of piedmont glaciers across glacial outwash during the LGM, which formed the physical boundary for some of the existing terminal moraines around the lakes within the CLD (Bentley, 1996; Bentley, 1997)."

Lines 617-619: 'The formation of ice-contact proglacial lakes likely occurred as a function of deglacial warming and ice retreat (Bentley, 1996)'. Well, kind of, but really it is due to mass loss combined with the underlying bed topography (restricting proglacial drainage). Not all glaciers have lakes, it is a product of retreating into their overdeepenings where meltwater becomes trapped. Re-phrase? E.g., examples outlined in recent papers by Dave Evans (Iceland).

We have adjusted the text following your recommendation: "The formation of ice-contact proglacial lakes likely occurred as a function of deglacial warming as ice retreated into overdeependings in the bedrock topography and filled with meltwater (Bentley, 1996)."

Lines 620-624: This doesn't really make sense: 'During deglaciation, iceberg calving into the proglacial lakes may have occurred (Bentley 1996,1997; Davies et al., 2020), with evidence suggesting that local topography and calving may have controlled the spatially irregular timing of abandonment from the terminal moraines surrounding the proglacial lakes (Bentley, 1997).
1) If you have an ice-contact lake, you have calving in the ablation area… re. '…iceberg calving into the proglacial lakes may have occurred'. This is not necessary; remove or re-phrase point.
2) Surely calving cannot control the glacier abandonment of the terminal moraine positions surrounding the proglacial lakes, because the glaciers had not yet developed lake? Am I missing something? Please clarify/re-write this point so that it is clear.

Thank you for pointing out this misrepresentation. We have adjusted the text following your recommendations:

"During deglaciation, proglacial lakes formed along the ice sheet margin (Bentley 1996,1997; Davies et al., 2020), with evidence suggesting that local topography and calving may have influenced the spatially varying retreat rates along these margins (Bentley, 1997)."

Lines 627-629: 'However, across our region, Heirman et al. (2011) indicate that it is not well constrained how the proglacial lakes in the CLD may have influenced local deglaciation, as more geomorphic data is needed'. It's true that we do not have geomorphological data, but a lot is probably at the base of these lakes (if at all) and would require bathymetric surveys. There has been some work done on glaciolacustrine varve sediments around Lago Buenos Aires (see: Bendle et al., 2017/2019 (?) papers) supports a period of rapid recession following the onset of calving as glaciers retreated and formed lakes in their overdeepenings. Maybe reference some of this work? It builds on these points.

Again, thank you for boosting our geomorphology knowledge for this region.  We have taken a look at the papers mentioned, and adjusted our text as follows.

"However, it is not well constrained how the proglacial lakes in the CLD may have influenced local deglaciation (Heirman et al., 2011).  While more geomorphic data is needed, recent work south of our study region (46.5°S) reconstructed early deglacial ice retreat using a glaciolacustrine varve record from Lago General Carrera-Buenos Aires (Bendle et al., 2019).  The authors find that following initial retreat due to deglacial warming, the ice margin retreated into the deepening proglacial lake which accelerated ice retreat in this region due to persistent calving, therefore supporting the role proglacial lakes likely played across the margins of the retreating PIS during the last deglaciation."

Figures:
Figure 1: Generally, I would make sure that maps have a north arrow, scale bar, and grid (latitude/longitude). Not necessary for all panels, but at least one to provide context.

We have added a scale bar and North arrow to Figure 2, which is shows our model domain across the CLD and bedrock geometry.  We hope this will add better context for the reader when considering our figures.

Figure 5: I am struggling to follow this description. Is it not dark blue where ice is persisting after 10 ka? Is something wrong here either in the caption or the figure? Please clarify and fix if needed.

The dark blue denotes deglaciation between 11 ka and 10 ka.  Our saturated color (<10 ka) is the gray color, for ice cover persisting <10 ka is assigned.

I hope that my comments are of some use to you and I look forward to seeing this published in the near future.

Thank you so much for your comments.  Together with the other reviews, we are grateful for the constructive feedback as it has surely made this a stronger paper.